

# A European giant: a large spinosaurid (Dinosauria: Theropoda) from the Vectis Formation (Wealden Group, Early Cretaceous), UK

Chris T. Barker[1,2], Jeremy A.F. Lockwood[3,4], Darren Naish[5], Sophie Brown[5], Amy Hart[5], Ethan Tulloch[5] and Neil J. Gostling[1,5]

[1] Institute for Life Sciences, University of Southampton, Southampton, United Kingdom
[2] Faculty of Engineering and Physical Sciences, University of Southampton, Southampton, United Kingdom
[3] School of Environment, Geography and Geosciences, University of Portsmouth, Portsmouth, United Kingdom
[4] Department of Earth Sciences, Natural History Museum, London, United Kingdom
[5] School of Biological Sciences, University of Southampton, Southampton, United Kingdom

Corresponding authors
Chris T. Barker, ctb1g14@soton.ac.uk
Neil J. Gostling,
n.j.gostling@soton.ac.uk

## ABSTRACT

Postcranial elements (cervical, sacral and caudal vertebrae, as well as ilium, rib and limb bone fragments) belonging to a gigantic tetanuran theropod were recovered from the basal unit (the White Rock Sandstone equivalent) of the Vectis Formation near Compton Chine, on the southwest coast of the Isle of Wight. These remains appear to pertain to the same individual, with enormous dimensions similar to those of the *Spinosaurus* holotype and exceeding those of the largest European theropods previously reported. A combination of features—including the presence of spinodiapophyseal webbing on an anterior caudal vertebra—suggest that this is a member of Spinosauridae, though a lack of convincing autapomorphies precludes the identification of a new taxon. Phylogenetic analysis supports spinosaurid affinities but we were unable to determine a more precise position within the clade weak support for a position within Spinosaurinae or an early-diverging position within Spinosauridae were found in some data runs. Bioerosion in the form of curved tubes is evident on several pieces, potentially related to harvesting behaviour by coleopteran bioeroders. This is the first spinosaurid reported from the Vectis Formation and the youngest British material referred to the clade. This Vectis Formation spinosaurid is unusual in that the majority of dinosaurs from the Lower Cretaceous units of the Wealden Supergroup are from the fluviolacustrine deposits of the underlying Barremian Wessex Formation. In contrast, the lagoonal facies of the upper Barremian–lower Aptian Vectis Formation only rarely yield dinosaur material. Our conclusions are in keeping with previous studies that emphasise western Europe as a pivotal region within spinosaurid origination and diversification.

## INTRODUCTION

The deposits of the internationally important Wessex Formation of the Isle of Wight—part of the Wealden Group (itself part of the Wealden Supergroup)—have been and remain exceptionally productive regarding dinosaur material and research (*Insole & Hutt, 1994*; *Radley & Allen, 2012c*; *Sweetman, 2011*). Indeed, the Wessex Formation has yielded almost all dinosaur fossils known from the Isle of Wight (*Martill & Naish, 2001b*). Its fluviolacustrine sediments preserve the remains of various tetanuran theropods, rebbachisaurid and titanosauriform sauropods, and a variety of ornithischians, including ankylosaurs and ornithopods (*Benton & Spencer, 1995*; *Lomax & Tamura, 2014*; *Martill & Naish, 2001a*; *Naish & Martill, 2007*; *Naish & Martill, 2008*). In contrast, dinosaur remains are rare in the overlying Vectis Formation (*Radley, Barker & Harding, 1998*), documented finds being limited to a handful of ornithopod, ankylosaur and indeterminate theropod specimens (*Benton & Spencer, 1995*; *Blows, 1987*; *Hooley, 1925*; *Martill & Naish, 2001a*; *Naish & Martill, 2008*; *Weishampel et al., 2004*; *White, 1921*). Ichnological remains referred to theropod, thyreophoran and ornithopod track-makers have also been reported from the Vectis Formation (*Pond et al., 2014*; *Radley, Barker & Harding, 1998*).

A number of large, fragmentary dinosaur bones, encased in a matrix matching the basal unit (the White Rock Sandstone) of the Vectis Formation, were found east of Compton Chine on the southwest coast of the Isle of Wight by Mr Nick Chase, Mr Mark Penn and Dr Jeremy Lockwood. The material was found loose in an intermittently exposed gutter located approximately 10 m from the cliff face, and its discovery here occurred over a period of several months. Taphonomic and anatomical evidence (discussed below) show that they belong to a single individual. Some of these bones were figured and alluded to in *Austen & Batten (2018)* but they have not previously been described. A list of character traits show that the specimen likely belongs to Spinosauridae and is thus the first member of this clade reported from the Vectis Formation. The specimen's large size is noteworthy and it appears to represent the largest theropod yet reported from the Wealden Supergroup and potentially from the European fossil record in general.

Our identification of this specimen as a spinosaurid is interesting in view of recent discoveries pertaining to spinosaurid diversity within the Wealden Supergroup. Spinosauridae is characterised by atypical cranial (and sometimes postcranial) morphologies indicative of divergent, semi-aquatic ecologies relative to related lineages (*Amiot et al., 2009*; *Amiot et al., 2010*; *Aureliano et al., 2018*; *Charig & Milner, 1997*; *Hassler et al., 2018*; *Ibrahim et al., 2020a*; *Ibrahim et al., 2014*; *McCurry et al., 2019*). Most studies support the division of Spinosauridae into Baryonychinae and Spinosaurinae (*Arden et al., 2019*; *Benson, 2010*; *Carrano, Benson & Sampson, 2012*; *Rauhut & Pol, 2019*; *Sereno et al., 1998*), although there are indications that support for this dichotomy may be weaker than customarily supposed (*Barker et al., 2021*; *Evers et al., 2015*). Most spinosaurids are from Early and mid Cretaceous strata but phylogenetic analyses support a Jurassic origin for the clade (*Barker et al., 2021*; *Carrano, Benson & Sampson, 2012*; *Hone & Holtz Jr 2017*) and isolated teeth suggest spinosaurid persistence into the Late Cretaceous (Santonian) (*Hone, Xu & Wang, 2010*).

To date, all formally published British spinosaurid remains come from the Berriasian–lower Aptian Wealden Supergroup, and include *Baryonyx walkeri* from the Upper Weald Clay Formation of the Weald sub-basin (*Charig & Milner, 1986*; *Charig & Milner, 1997*), and *Ceratosuchops inferodios* and *Riparovenator milnerae* from the Wessex Formation of the Wessex sub-basin (*Barker et al., 2021*). Additional fragmentary material has been recovered throughout the Wealden succession (*Buffetaut, 2010*; *Charig & Milner, 1997*; *Hutt & Newbery, 2004*; *Martill & Hutt, 1996*; *Milner, 2003*; *Naish, 2011*; *Naish, Hutt & Martill, 2001*; *Salisbury & Naish, 2011*; *Turmine-Juhel et al., 2019*). This Wealden Supergroup material pertains exclusively to Baryonychinae and spinosaurines are currently unknown from the British fossil record. This contrasts with equivalent strata in Iberia, where evidence of both clades is known (see *Malafaia et al. (2020a)* for a review of the Iberian spinosaurid record).

In the present contribution, we provide osteological descriptions and comparisons of the better-preserved remains (several additional fragments, including some large pieces, could not be readily identified but are briefly reported in the supplementary information), and include the ''White Rock spinosaurid'' in a phylogenetic analysis in order to further test its affinities. We also remark upon the biostratinomic context of these finds, and briefly describe the bioerosion apparent on several elements.

## Geological context

The Wealden Supergroup of southern England is a succession of largely non-marine strata accumulated during the Early Cretaceous (late Berriasian–early Aptian) and mainly deposited in two sub-basins (Fig. 1A): the larger Weald sub-basin of south-eastern England, and the smaller Wessex sub-basin of the Isle of Wight and central-southern England (*Batten, 2011*; *Radley & Allen, 2012a*).

Within the latter, the succession consists of the younger Wealden Group and older Purbeck Limestone Group. The Wealden Group on the Isle of Wight (Fig. 1B) predominantly crops out along the island's southwest coast, with a smaller exposure occurring along the southeast coast. Both areas reveal the entirely Barremian and predominately alluvial facies of the Wessex Formation (deposited in a fluviolacustrine setting) as well as the overlying late Barremian–early Aptian Vectis Formation (*Radley & Allen, 2012c*; *Sweetman, 2011*) (Fig. 1C).

The three constituent members of the 67 m thick Vectis Formation represent the return to coastal lagoonal environments that occurred prior to the Aptian marine transgression and are characterised by low diversity ostracod and mollusc assemblages (*Radley, Barker & Harding, 1998*; *Ruffell, 1988*; *Sweetman, 2011*). The largely argillaceous Cowleaze Chine and Shepherd's Chine members form the base and top of the formation respectively, denoting low-energy subaqueous or mudflat environments. The Barremian–Aptian boundary occurs within the Shepherd's Chine Member (*Kerth & Hailwood, 1988*; *Robinson & Hesselbo, 2004*). The interposing Barnes High Sandstone Member represents deltaic inundation into the lagoon (*Radley, Barker & Harding, 1998*).

At the Atherfield type locality and extending west of Cowleaze Chine, a pale, metre-thick sandstone unit in-fills the ''dinoturbated'' uppermost stratum (the *Hypsilophodon* bed)

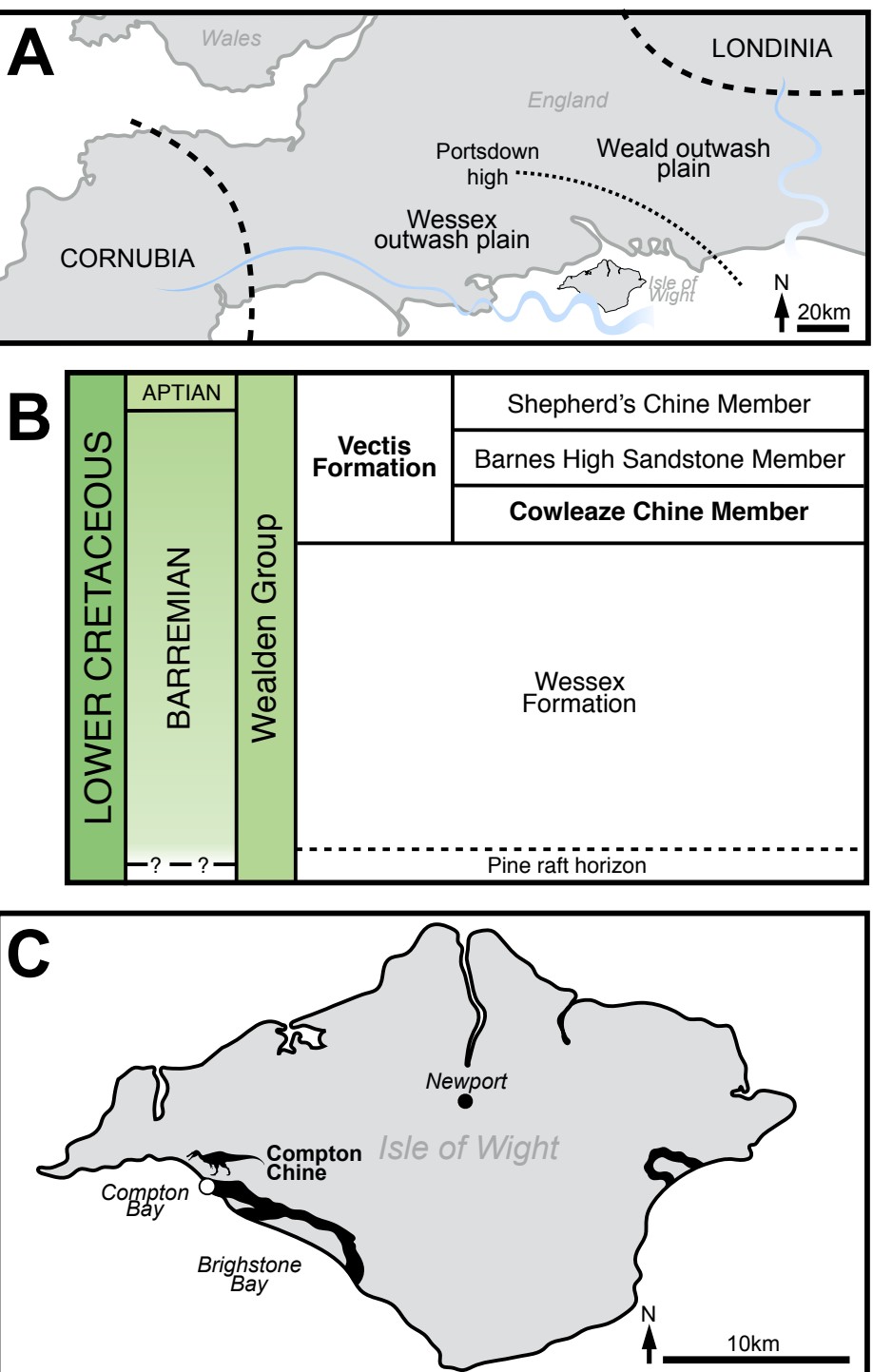

**Figure 1** **General geological context of the White Rock spinosaurid material.** (A) Schematic palaeogeographic map of the Wealden Supergroup, highlighting the Wessex and Weald sub-basins (from *Barker et al. (2021)*, modified from *Penn et al. (2020)*: Fig. 2). (B) Schematic stratigraphy of the Wealden Group on the Isle of Wight (modified from *Radley & Allen (2012c)*: Fig. 6), with relevant strata highlighted. (C) Map of the Isle of Wight, highlighting the outcrops of the Vectis Fm. and location of the spinosaurid remains (modified from *Ruffell (1988)*: Fig. 1). Spinosaurid silhouette courtesy of Dan Folkes (CC-BY 4.0).

of the underlying Wessex Formation and forms the base of the Cowleaze Chine Member (*Radley, Barker & Harding, 1998*; *Sweetman, 2011*). Known as the White Rock Sandstone, it is interpreted as narrow fluvial channels intersecting a marginal lagoonal sand-flat deposit laid down by climatically-controlled terrestrial runoff and intermittent lagoonal influxes (*Radley, Barker & Harding, 1998*; *Sweetman, 2011*). The lower part of the White Rock Sandstone is formed of laminated, cross-laminated or burrow-mottled sandstone (*Radley, Barker & Harding, 1998*). Lenses of fusain-rich carbonaceous sandstone, organic-rich mudstones, and poorly sorted conglomerate are interspersed throughout this lower part; the conglomerates occasionally yield worn reptilian bone fragments (*Radley, Barker & Harding, 1998*).

Due to a fault, the Vectis Formation crops out at two sites in Compton Bay, the larger exposure being located to the east near Shippards Chine and the other towards the west, nearer Compton Chine (Fig. 2A). The specimens were all found in front of the ∼34 m thick (*Radley & Allen, 2012c*) more westerly exposure, along an approximately 50 m stretch of foreshore. Here, the basal ∼60 cm unit of the Vectis Formation is lithologically variable and includes a fine sandstone and a pale jarositic siltstone, resembling the higher part of the White Rock Sandstone at the previously described type locality, and is marked at the outcrop by a line of water seepage (*Radley & Barker, 1998*). This White Rock Sandstone equivalent forms an obvious layer that is distinct from the dark grey mud and siltstones of the lagoonal sediments of the Cowleaze Chine member and the varicoloured palaeosols or grey plant debris beds of the Wessex Formation (Fig. 2B). Although all the spinosaurid specimens reported here were found on the foreshore, adhering matrix closely matches that of the White Rock Sandstone equivalent in all specimens, and the remains were likely present on the foreshore due to a cliff fall (though the possibility remains that their presence is due to erosion through a wave cut platform) (Fig. 2C). Generally, the White Rock equivalent at this location contains few macroscopic fossils except for sporadic fragments of fusain and bone. Ichnites are represented by the occasional gastrolith and infrequent burrows usually ∼1 cm in diameter.

## METHODS

### Measurements
Measurements were taken in millimetres using digital callipers and rounded to one decimal point.

### Terminology
Nomenclature of the vertebral neural arch fossae and laminae follows *Wilson et al. (2011)*. Relative position within the axial series is based on the suggestions of *Evers et al. (2015)* and we also follow the latter authors in their repositioning of the *Baryonyx walkeri* type presacral series. Nomenclature of the various ichnological features found on these specimens follows the ichnotaxobases provided by *Pirrone, Buatois & Bromley (2014)*.

### Phylogenetic analysis
The White Rock spinosaurid was included in a comprehensive phylogenetic matrix derived from *Cau (2018)* and implemented in *Barker et al. (2021)*, focusing on non-coelurosaurian

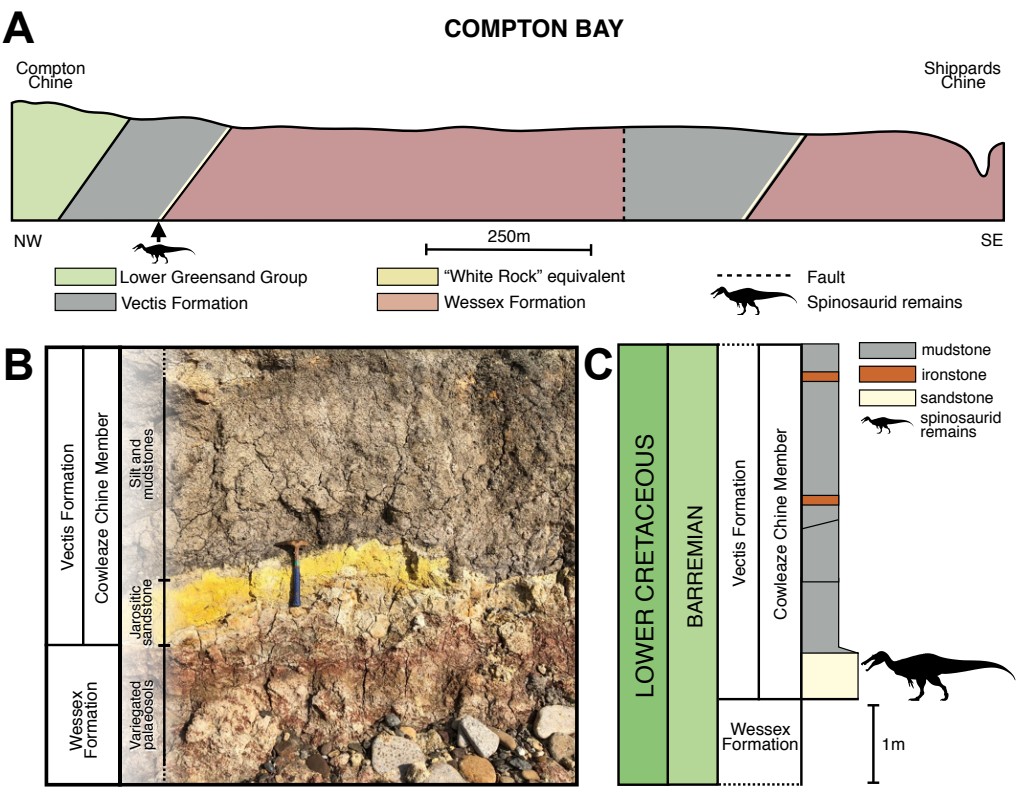

**Figure 2** **Stratigraphic context of the White Rock spinosaurid material.** (A) View of the cliff between Compton Chine and Shippards Chine (Compton Bay), highlighting the members of the Wealden Group and overlying Lower Greensand Group (from *Radley & Barker, 1998)*: Fig. 2). (B) Junction between the Wessex and Vectis formations located towards Compton Chine. (C) Vertical section through the lower unit of the Vectis Formation, Compton Bay, Isle of Wight (modified from *Radley & Allen (2012c)*: Fig. 26). Spinosaurid silhouette courtesy of Dan Folkes (CC-BY 4.0).

tetanurans. Following our positional identifications (see "Descriptive osteology"), IWCMS 2018.30.1 was scored as an anterior dorsal vertebra, whilst IWCMS 2018.30.3 was scored as an anterior caudal vertebra.

Scores for five character statements concerning the caudal vertebrae of the two operational taxonomic units (OTUs) *Baryonyx* (NHMUK PV R 9951) and *Riparovenator* (IWCMS 2020.447.1, 2) were changed relative to the analysis in *Barker et al. (2021)*. For *Baryonyx*, these changes related to the caudal neural arch characters (Ch.) 358, 359, 868 and 1576. An isolated neural arch belonging to NHMUK PV R 9951 was identified as that of an anterior caudal vertebra by *Charig & Milner (1997)*. However, the presence of a hyposphene and well-developed centrodiapophyseal laminae alternatively suggest that the element instead belongs to a posterior dorsal vertebra, an identification also proposed by *Charig & Milner (1997)*. Given this uncertainty, we opt to re-code the above character as "?". Regarding *Riparovenator*, Ch. 1035 (originally Ch. 99 of *Carrano & Sampson (2008)* and concerning caudal neural spine morphology) was mis-scored and has been changed to state 1 to reflect their abbreviated state. All other scores and "OTUs" remained the same

as in the *Barker et al. (2021)* analysis, although we acknowledge the recent designation of the specimen ML 1190 as the holotype of the new spinosaurid taxon *Iberospinus natarioi* (*Mateus & Estraviz-López, 2022*), which also includes some fragmentary new material.

The final matrix contains 41 operational taxonomic units coded for 1810 binary character statements. The analysis was performed in TNT v1.5 (*Goloboff & Catalano, 2016*). A driven search using 100 initial addition sequences was performed *via* the "New Technology Search" function, with default settings employed for sectorial, ratchet, drift and fusion. Tree islands were further explored *via* a round of tree bisection and reconnection (TBR) using the "Traditional search" function, and results were initially explored *via* a strict consensus. Improved resolution was achieved *via* the identification of wildcard OTUs using the iterPCR method (*Pol & Escapa, 2009*) implemented in TNT (*Trees > Comparisons > Iter PCR*). A reduced consensus tree was calculated following the pruning of these OTUs.

Nodal support was assessed *via* Bremer (decay indices; *Trees > Bremer*) and jackknife (*Analyze > Resampling*) values. The former were obtained for the strict consensus by retaining trees suboptimal by 10 steps, whilst those of the reduced consensus were calculated using existing suboptimal trees with the exclusion of the wildcard OTUs identified previously. Jackknife values were calculated using 1000 pseudoreplicates under a "traditional search" function, also excluding *a priori* the wildcard OTUs. We report both absolute and GC frequency values.

## RESULTS
### Systematic palaeontology

DINOSAURIA *Owen, 1842*
THEROPODA *Marsh, 1881*
TETANURAE *Gauthier, 1986*
SPINOSAURIDAE *Stromer, 1915*

Spinosauridae indet.
*Referred specimens:* IWCMS 2018.30, which includes a probable yet fragmentary anterior dorsal vertebra (2018.30.1), a pair of fused sacral centra (2018.30.2), a partial anterior caudal vertebra (2018.30.3), a sacrocaudal centrum fragment (2018.30.4), rib fragments (2018.30.5, 6), pieces of ilium (2018.30.7, 8) and portions of long bone (2018.30.9, 10). Several other indeterminate fragments have also been recovered (see also supplementary information).
*Locality and Horizon:* White Rock Sandstone equivalent, Compton Chine, Vectis Formation (late Barremian).

### Descriptive osteology
#### Axial elements
*IWCMS 2018.30.1 (Anterior dorsal vertebra).* This element is represented by the majority of the centrum and a portion of the right neural arch (Fig. 3), metric data of which

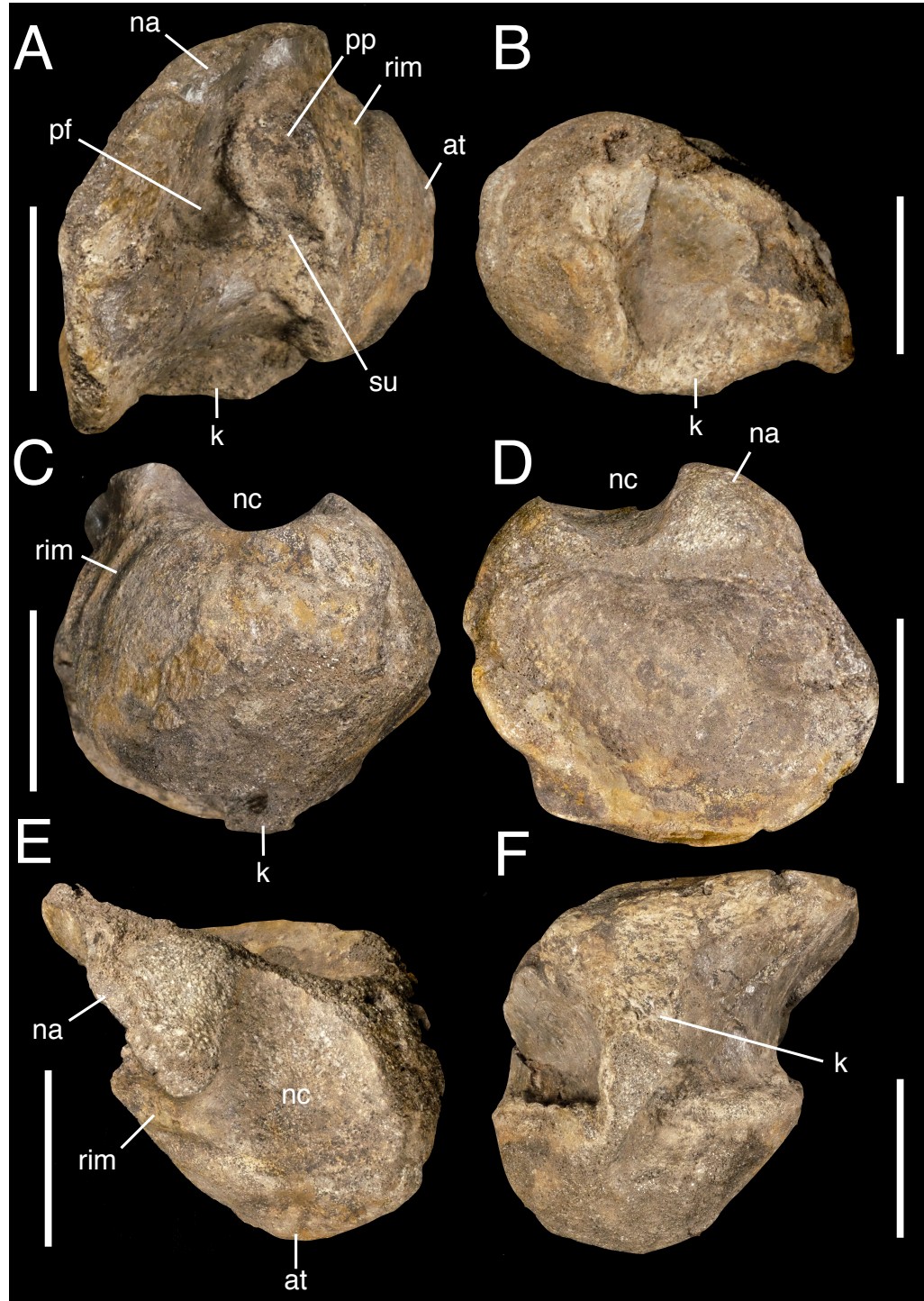

**Figure 3  Anterior dorsal vertebral fragment IWCMS 2018.30.1.** In (A) right lateral, (B) left lateral, (C) anterior, (D) posterior, (E) dorsal and (F) ventral views. Abbreviations: at, anterior tuberosity; k, keel; na, neural arch; nc, neural canal; pf, pneumatic foramen; pp, parapophysis; rim, flattened rim around the anterior articular facet; su, sulcus. Scale bar: 50 mm.

**Table 1  Metric data for IWCMS 2018.30.1.** An asterisk (*) denotes taphonomic damage. Measurements are in millimetres (mm).

| | |
|---|---|
| Anteroposterior length of the centrum (between ventral rims)* | 69.4 |
| Dorsoventral midline height of the anterior articular facet* | 75.3 |
| Mediolateral width of the anterior articular facet* | 99.2 |
| Dorsoventral midline height of the posterior facet* | 92.5 |
| Mediolateral width of the posterior facet* | 118.5 |
| Dorsoventral height of the right parapophysis | 27.8 |
| Anteroposterior length of the right parapophysis | 25.7 |
| Mediolateral width of the neural canal | 39.7 |

are presented in Table 1. The left side of the anterior and posterior articular facets are substantially abraded, as is the ventral rim of the anterior facet, exposing cancellous bone and its trabeculae; this ventral abrasion has also affected the anterior part of the ventral keel. A sub-circular portion of the bone has been lost from the right ventral surface, including a part of the ventral keel. The extensive damage to the neural arch and loss of most of its structures has also exposed cancellous bone across the dorsal surface, as well as on the floor of the wide neural canal. The specimen has likely experienced some plastic deformation; given the posterolaterally facing rather than laterally facing parapophysis, this deformation may be related to compressive forces.

The anteroposteriorly abbreviated centrum is opisthocoelous, with a pronounced anterior convexity and posterior concavity. The nature of the neurocentral suture is ambiguous; a suture-like feature is visible in anterior and right lateral view and located above the parapophysis, suggesting the latter is thus entirely centrum-bound if genuine. However, this structure may be a taphonomic artefact and not a suture at all.

Both articular facets are mediolaterally wide and in line with one another (*i.e.,* the anterior facet is not dorsally offset relative to the posterior facet); the posterior facet protrudes lateral to the extremities of the anterior equivalent when the specimen is viewed dorsally. The anterior facet lacks any notable inclination but is not uniformly convex since a subtle, median tuberosity is present. This tuberosity is visible in lateral view and protrudes a short distance anteriorly (Fig. 3A). The dorsal margin of the anterior facet is subtly concave dorsal to the tuberosity, such that the dorsal margin is indented in anterior view. A distinct flattened rim is present on the undamaged dorsal portion of the right side of the facet, demarcated posteriorly by a low ridge.

The concave right lateral surface possesses a sediment-filled pneumatic foramen, located posteroventral to the ipsilateral parapophysis. The original shape of the foramen cannot be ascertained, and damage precludes identification of the foramen on the left side. The foramen appears to communicate with a shallow yet broad sulcus that cuts into the centrum ventral to the parapophysis (Fig. 3A). The parapophysis is sub-circular and largely flattened.

Ventrally, the centrum possesses a stout keel, which is better developed anteriorly. A ventral fossa on the left side of the centrum contributes somewhat to the keel's pronounced
nature, although this is not mirrored on the right. The posterior portion of the keel expands mediolaterally as it becomes confluent with the posterior articular margin.

Regarding its position within the axial series, the anterodorsal location of the parapophysis, sub-parallel (rather than offset) relationship between the articular facets, and possession of a prominent ventral keel (*Evers et al., 2015*) suggest an anterior dorsal position for IWCMS 2018.30.1. Tetanuran parapophyses typically migrate onto the neural arch between the 2nd and 7th dorsal (*Holtz, Molnar & Currie, 2004*). It is unclear whether the parapophysis remains restricted to the centrum in the specimen discussed here but its position is most similar to that present in the second dorsal vertebrae of *Baryonyx* (NHMUK PV R9951; fourth dorsal of *Charig & Milner (1997)*) and second and third dorsals of cf. *Suchomimus* (MNBH GAD70, *Ibrahim et al. (2020b)*: Figure 130). Accordingly, we identify IWCMS 2018.30.1 as a second or third dorsal vertebra.

*IWCMS 2018.30.2 (Sacral vertebrae).* Two sacral centra, fused at their intercentral junction, are known (Fig. 4): the centra are relatively well preserved, but the neural arches and sacral ribs are missing. The only breakage consists of shallow cracks on the smooth external surfaces of the centra, and a large oblique transverse crack near the posterior articular facet of the more posterior centrum. Abrasion has damaged several surfaces to some extent, but most notably affects the sacral rib attachments as well as both articular facet rims and the conjoined intercentral junction, where the underlying trabeculae are exposed. With regard to abrasion of the exposed anterior and posterior facets, the external bone in the more anterior centrum is largely intact in anterior view, whereas abrasion of the facet rims is more extensive in the posterior element when it is viewed posteriorly (the central portion of this facet is nonetheless preserved). An indeterminate mass of bone and matrix is cemented onto the floor of the neural canal of the more posterior centrum. Metric data are presented in Table 2.

The robust centra are longer than tall, and are approximately in line with one another. The exposed hemielliptical anterior facet of the anterior element is flat and notably larger than the sub-circular posterior facet of the more posterior element. The latter appears convex, although this is likely due to abrasion of the facet's rim.

The sacral rib attachments are large, subtriangular and located anterodorsally on the lateral surfaces of the centra. They are asymmetrical in the anterior element, and the right attachment facet appears larger and more prominent. On the posterior centrum, the sacral rib attachments appear less developed, although it seems likely they have been substantially weathered. The floors of the intervertebral foramina are visible bilaterally as wide and posteroventrally trending channels present on the dorsal surface of the more posterior centrum.

The dorsolateral surfaces, ventral to the neurocentral junction, are variably indented. The right lateral depression on the anterior centrum is best developed, in contrast to its far shallower counterpart, whilst those on the posterior centrum are more similar in development. These depressions do not house pneumatic foramina, and their poor development indicates these are unlikely to pertain to a pneumatic system.

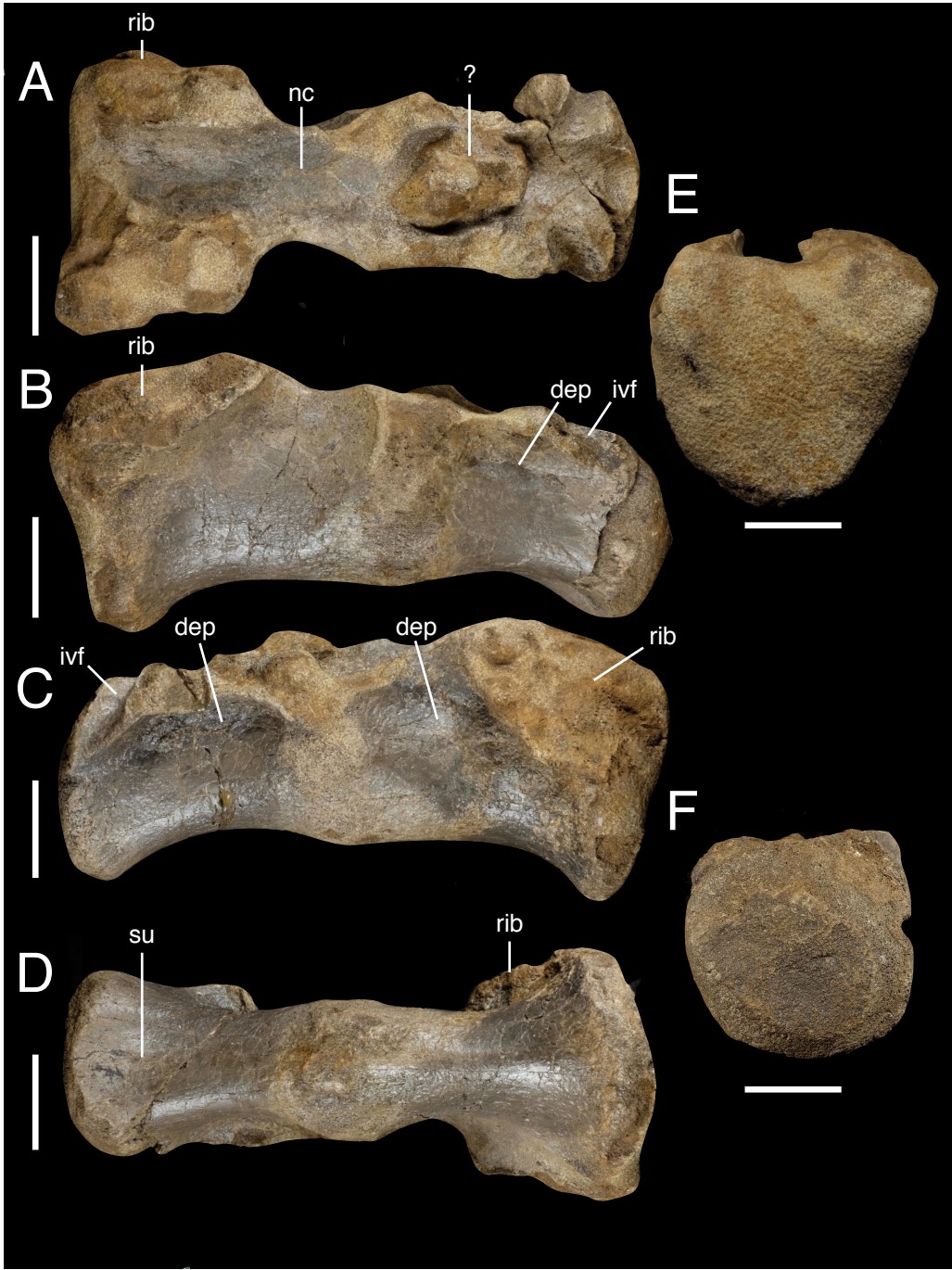

**Figure 4 Conjoined sacral centra IWCMS 2018.30.2.** In (A) dorsal, (B) right lateral, (C) left lateral, (D) ventral, (E) anterior and (F) posterior views. Abbreviations: dep, depression; ivf, floor of the intervertebral foramen; nc, neural canal; rib, sacral rib attachment; su, sulcus. Scale bars: 50 mm.

**Table 2** **Metric data for IWCMS 2018.30.2.** An asterisk (*) denotes taphonomic damage. Measurements are in millimetres (mm).

| | |
|---|---|
| Maximum anteroposterior length of the conjoined centra | 298 |
| Anteroposterior length of anterior centrum | ~156 |
| Anteroposterior length of posterior centrum | ~142 |
| Dorsoventral midline height of the exposed anterior articular facet* | 118.1 |
| Mediolateral midline width of the exposed anterior articular facet* | 126.2 |
| Dorsoventral midline height of the exposed posterior facet* | 107.9 |
| Mediolateral width of the exposed posterior facet* | 102.7 |
| Mediolateral width of the neural canal | 40.7 |

The ventral margins are only shallowly concave in lateral view. The ventral surface of the anterior centrum is rounded in transverse section along its length. Similar rounding is present on the posterior centrum; however, this element goes on to develop a shallow midline sulcus posteriorly. This sulcus is associated with a degree of mediolateral expansion of the bone, with the latter centrum thus appearing posteriorly wider relative to the equivalent end of the anterior element when viewed ventrally.

The relative position of the sacral vertebrae is difficult to determine given their incompleteness, and it is perhaps unusual that elements of this size are not more extensively fused to other sacral elements. The plesiomorphic dinosaurian (and archosaurian) sacrum consisted of two "primordial vertebrae" (*Langer & Benton, 2006*; *Moro et al., 2021*). This count increased to five in tetanurans *via* the addition of dorso- and caudosacrals (*Holtz, Molnar & Currie, 2004*). The primordial sacral vertebrae are thought to fuse prior to the evolutionarily 'younger' elements (*O'Connor, 2007*), suggesting that IWCMS 2018.30.2 may represent this pair in the absence of a completely fused series. However, recognition of sacral fusion patterns in theropods remain complicated (*Moro et al., 2021*) and the identification of primordial sacrals is largely based on their sacral ribs and associated attachment points on the ilium (*Nesbitt, 2011*), neither of which can be assessed here.

*IWCMS 2018.30.3 (Anterior caudal vertebra).*  A large partial caudal vertebra preserves only its posterior portion, having suffered a transverse shear posterior to the prezygapophyses (Fig. 5). It is among the most complete and informative of the elements known for this dinosaur. Fine cracks are apparent across the external bone surfaces, most notably affecting the centra. Both transverse processes and the neural spine have been lost, whilst abrasion to the postzygapophyses and margins of various neural arch laminae is apparent. Minor crushing appears to affect the left side of the element, as evidenced by the flattening of the ipsilateral rim of the posterior articular facet in posterior view. The left portion of said facet also appears abraded such that the underlying trabecular bone is exposed; abrasion also affects the rim of the right half of the facet. Metric data are presented in Table 3.

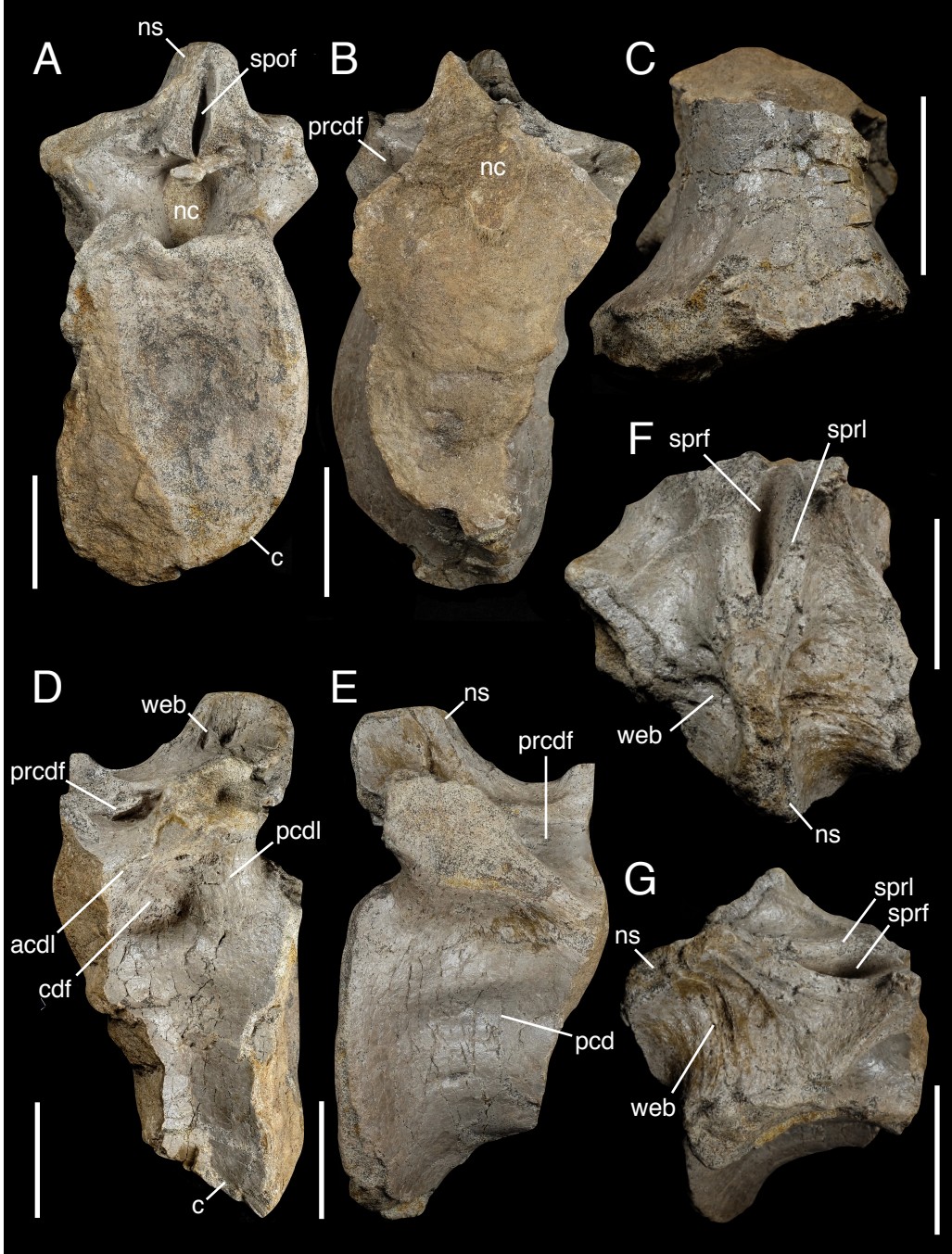

**Figure 5** **Partial anterior caudal vertebra IWCMS 2018.30.3.** In (A) posterior, (B) anterior, (C) ventral, (D) left lateral, (E) right lateral, (F) dorsal and (G) right dorsolateral oblique views. Abbreviations: acdl, anterior centrodiapophyseal lamina; c, centrum; cdf, centrodiapophyseal fossa; nc, neural canal; ns, neural spine; pcd, pleurocentral depression; pcdl, posterior centrodiapophyseal lamina; prcdf, prezygocentrodiapophyseal fossa; sprf, spinoprezygapophyseal fossa; sprl, spinoprezygapophyseal lamina; spof, spinopostzygapophyseal fossa; web, spinodiapophyseal webbing. Scale bars: 50 mm.

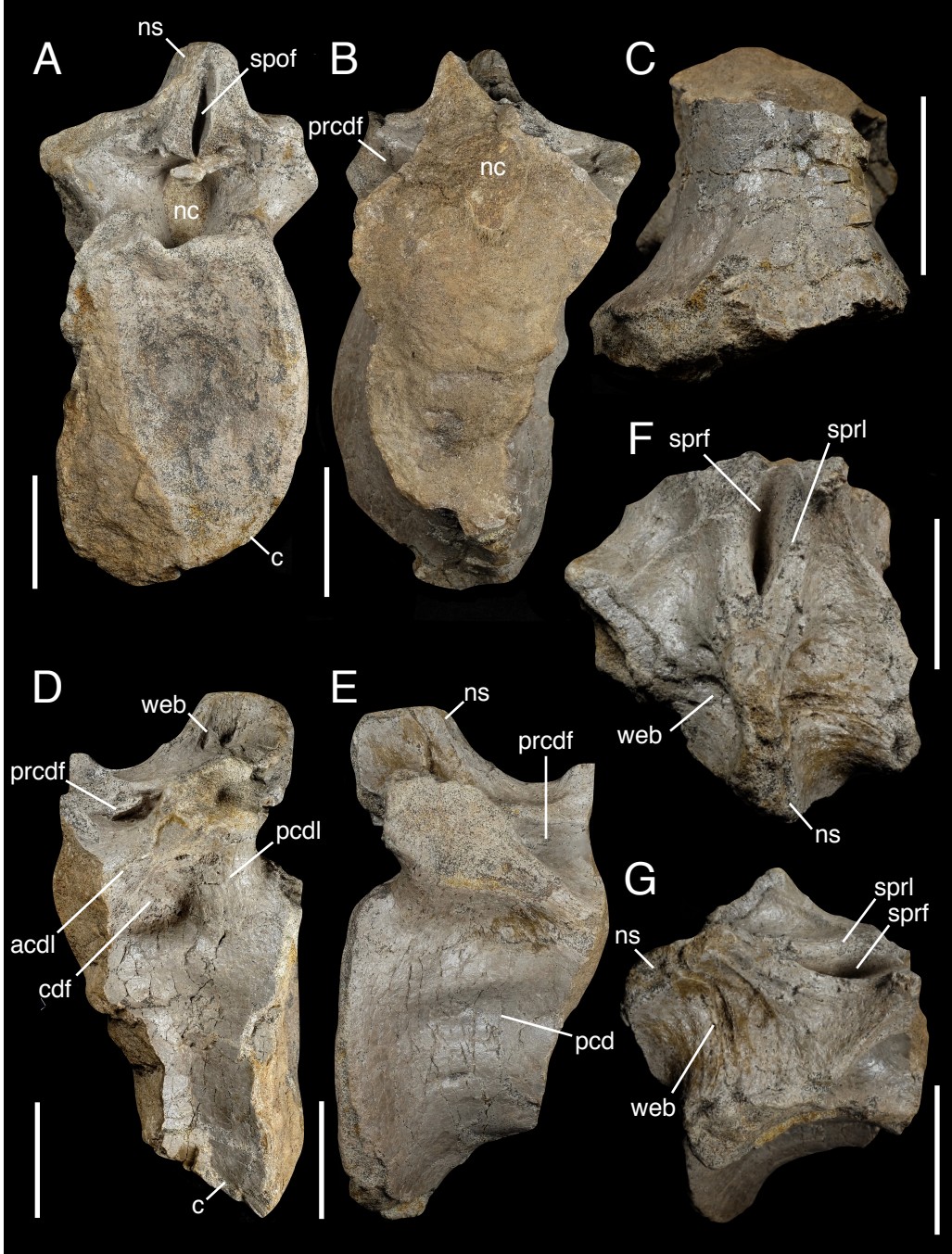

**Table 3 Metric data for IWCMS 2018.30.3.** An asterisk (*) denotes taphonomic damage. Measurements are in millimetres (mm).

| | |
|---|---|
| Dorsoventral height of posterior articular facet | 159.8 |
| Mediolateral width of the posterior articular facet* | 112.8 |
| Anteroposterior depth of the concavity of the posterior articular facet* | 25.3 |
| Anteroposterior length of the preserved centrum (right side) | 106.5 |
| Dorsoventral height of the anterior neural canal | 38.6 |
| Mediolateral width of the anterior neural canal | 29.5 |
| Anteroposterior length of the base of the neural spine | 49.6 |
| Mediolateral width of the base of the neural spine | 16.8 |

In life, the centrum was tall relative to its width (Fig. 5A), with the dorsoventral midline height of the posterior facet appearing unaffected by the crushing experienced along its left lateral side. The lateral margins are concave in coronal section, as is the ventral margin in lateral view. It is difficult to determine whether the neurocentral suture is closed: in places, the suture looks highlighted by specks of a black mineral (which also dots many of the abraded surfaces and cracks throughout the element), but it is unclear if this represents retention of the open state or is a taphonomic artefact. The broken anterior surface does not preserve obvious evidence of internal pneumatic features such as camerae or camellae (*Britt, 1993*; *Britt, 1997*) (Fig. 5B). The distinction between the cortical and cancellous bone is obvious in places, with the former measuring 4.8 mm on the left ventrolateral side; it appears to thin dorsally towards the neurocentral suture. The cross-section of the infilled neural canal is visible in anterior view. It is largely circular, but its mid-ventral margin bulges ventrally.

The ventral surface of the centrum is heavily distorted. Although no keel is present, crushing on the left side has distorted the surface and its original shape can only be supposed; based on the better-preserved right side, it was likely largely convex in transverse section (Fig. 5C).

The lateral surfaces of the centrum present an elongate pleurocentral depression dorsally. On the better-preserved right side, a trifecta of small and presumably vascular foramina penetrate the right lateral surface. The dorsal two are smaller and located along the anterior and posterior ventral margins of the pleurocentral depression, with the larger, more ventral foramen positioned in line with the latter. Posteriorly, the mid-dorsal rim of the tall and moderately concave posterior articular facet is shallowly indented, above which sits the inversely ovate neural canal.

The neural arch is robust, with thick walls made visible in the anterior cross-section. It preserves various fossae, some of which are delimited by stout laminae and may bilaterally vary in shape (Figs. 5D–5F). Along the anterodorsal midline, the spinoprezygapophyseal fossa is deepest posteriorly and narrows mediolaterally towards the neural spine, being bordered by variably developed laminae; the right lamina is sharper than the contralateral structure. The dorsal rim of the former lamina is more complete, preserving a dorsally
curving anterior portion where it rose to meet the ipsilateral prezygapophyseal pedicle in lateral view.

Prezygocentrodiapophyseal and centrodiapophyseal fossae excavate the lateral neural arch surfaces. The former are deep and possess a largely triangular outline *via* two constraining laminae: the largely horizontal prezygodiapophyseal lamina forms its dorsal border, while the notably thick and obliquely oriented anterior centrodiapophyseal lamina delimits the fossa ventrally. The latter also forms the anterior margin of the bilaterally asymmetrical centrodiapophyseal fossae. The left is more developed, excavating the neural arch ventral to the transverse process to a deeper extent; the right, fossa, in contrast, is hardly perceptible. Posteriorly, the posterior centrodiapophyseal lamina forms a thick buttress to the transverse process. What remains of the transverse processes indicate these were massively constructed and possibly posterolaterally projecting. Postzygocentrodiapophyseal fossae are absent in this element.

The neural spine is posteriorly positioned on the neural arch. The base of the spine is mediolaterally thin and anteroposteriorly short. It is bilaterally webbed *via* variably developed spinodiapophyseal sulci and ridges (Figs. 5F, 5G). The postzygapophyses are insufficiently preserved at their posterior ends to warrant useful description, although the dorsoventrally tall spinopostzygapophyseal fossa they enclosed is narrow and slit-like. No obvious hyposphene is present ventral to the remnants of the postzygapophyses (indeed, there appears to be no space between the spinopostzygapophyseal fossa and dorsal margin of the neural canal in which one could be present), although a small mass of cemented bone and sandstone overhangs the neural canal posteriorly.

The positioning of IWCMS 2018.30.3 within the caudal series derives from multiple lines of evidence. Indeed, several more anterior axial positions can be readily excluded. The dorsal positions of the transverse processes and their buttressing laminae eliminate most of the cervical series from consideration. In addition, the absence of a ventral keel is inconsistent with the condition present in posterior cervicals and anterior dorsals. The absence of internal pneumaticity within the centrum also indicates a more posterior position given that pneumatisation of the cervical and anterior dorsal centra is the "common pattern" amongst theropods (*Benson et al., 2012*). The lack of sacral ribs or their facets excludes a sacral position. Finally, the ovate shape of the posterior articular facet resembles the condition present in theropod posterior dorsal and anterior caudal vertebrae (*Rauhut, 2003*), as does the presence of spinodiapophyseal webbing (observed in such elements in spinosaurid taxa especially).

We consider it most likely that IWCMS 2018.30.3 represents an anterior caudal vertebra, rather than the mid- or posterior dorsal vertebra for several reasons: a hyposphene, postzygocentrodiapophyseal fossae and accessory centrodiapophyseal laminae are all absent, and the neural spine is anteroposteriorly short. Hyposphenes are typical of dorsal vertebrae in large saurischians (although they can occur in the posterior cervical and anterior caudal vertebrae too) (*Langer, 2004*; *Rauhut, 2003*; *Stefanic & Nesbitt, 2019*), and are present in the mid- and posterior dorsal vertebrae of *Baryonyx* (NHMUK PV R9951) (*Charig & Milner, 1997*), IWCMS 2012.563 (*Hutt & Newbery, 2004*), *Suchomimus* (MNN GDF 500) and *Ichthyovenator* (MDS BK 10-01) (*Allain et al., 2012*) where they are ventral

to a broad spinopostzygapophyseal fossa and separate the latter from the neural canal. Hyposphene-free anterior caudal vertebrae are common amongst spinosaurids (*Barker et al., 2021*): a hyposphene is present in the putative anterior caudal neural arch of *Baryonyx* (*Charig & Milner, 1997*) but—as discussed above—the identification of this element as an anterior caudal vertebra may be an error. The absence of a hyposphene means that the spinopostzygapophyseal fossa is located dorsal to the neural canal (as seen in IWCMS 2018.30.3). The fossae concerned may also be narrower than their equivalents in the dorsal vertebrae, as noted in the anterior caudal vertebrae of *Riparovenator* (*Barker et al., 2021*) and *Vallibonavenatrix* (*Malafaia et al., 2020b*), although we concede that the narrow condition present in IWCMS 2018.30.3 may be exaggerated by loss of its postzygapophyses.

The pair of centrodiapophyseal fossae in IWCMS 2018.30.3 also differs from the three present in the mid and posterior dorsal vertebrae of such spinosaurids as *Baryonyx* (*Charig & Milner, 1997*), *Ichthyovenator* (*Allain et al., 2012*), *Vallibonavenatrix* (*Malafaia et al., 2020b*), *Spinosaurus* (*Stromer, 1915*) and *Suchomimus* (MNN GDF 500). Some of these taxa present an accessory centrodiapophyseal lamina in this vicinity, a trait typically recovered as synapomorphic of Baryonychinae but also present in the phylogenetically labile taxon *Ichthyovenator* (*Allain et al., 2012*; *Barker et al., 2021*; *Benson, 2010*; *Carrano, Benson & Sampson, 2012*; *Holtz, Molnar & Currie, 2004*; *Rauhut & Pol, 2019*). Given the absence to date of spinosaurine spinosaurids (see also below) in the Wealden Supergroup, an accessory lamina might be expected if this element were a mid- or posterior dorsal vertebra.

The lack of a chevron facet—a characteristic feature of caudal vertebrae—would appear to count against a caudal identification for IWCMS 2018.30.3. However, chevron facets are absent on the anteriormost caudal centra of some tetanurans (*Holtz, Molnar & Currie, 2004*). Further support for a caudal identification is provided by the anteroposteriorly short and posteriorly positioned neural spine, the position and anatomy of which recalls the condition in the anterior caudal vertebrae of *Riparovenator* (*Barker et al., 2021*) (see also Table 4). Caudal vertebrae of basal tetanurans may be amphicoelous or amphiplatyan (*Holtz, Molnar & Currie, 2004*), and the concave posterior facet of IWCMS 2018.30.3 recalls the amphicoelous anatomy of *Spinosaurus* (*Stromer, 1915*), *Ichthyovenator* (*Allain et al., 2012*), the spinosaurine FSAC-KK 11888 (*Ibrahim et al., 2020a*) and *Vallibonavenatrix* (*Malafaia et al., 2020b*).

*IWCMS 2018.30.4 (Sacrocaudal fragment).* The damaged and fragmentary vertebra (Figs. 6A–6D) was also recovered; it lacks many of its original margins and its dorsal surface is obscured by matrix. Useful morphometric data is difficult to obtain in light of its preservation. Its asymmetry presumably represents a degree of plastic deformation. The anterior and posterior surfaces have been damaged, although one surface (perhaps the posterior one, see below) appears to preserve a degree of bevelling in its ventral part, though this may be taphonomic in origin. The fragment possesses a width of 68.1 mm (measured across the ventral midpoint), a maximum height of 70.6 mm, and a maximum length of 74.6 mm. The most noteworthy osteological feature pertains to a prominent and wide anteroposteriorly oriented sulcus on its ventral surface.

Barker et al. (2022), *PeerJ*, DOI 10.7717/peerj.13543

**Table 4 Size of the anterior caudal neural spine base (collected from the most anterior preserved caudal element) relative to their respective neural arch in select spinosaurids.** Note that data for key taxa (*e.g., Baryonyx* and *Suchomimus*) is missing due to preservation. An asterisk (*) denotes minimum metric due to preservation. Where neural arch base lengths are unknown, centrum length is used (denoted by †). Data collected from *Allain et al. (2012)*, *Ibrahim et al. (2020a)* and *Samathi, Sander & Chanthasit (2021)*. *Riparovenator* and FSAC-KK 11888 calculated via images using the scale function in FIJI (*Schindelin et al., 2012*).

| Specimen | Spinosauridae indet. (IWCMS 2018.30.3) | "Phuwiang spinosaurid B" (SM-PW9B-15) | *Riparovenator* (IWCMS 2020.447.3) | *Ichthyovenator* (MDS BK10-02) | Spinosaurinae indet. (FSAC-KK 11888) |
|---|---|---|---|---|---|
| Basal neural arch length (mm) | 112.9* | 69 | ~138 | 101† | ~55 |
| Basal neural spine length (mm) | 49.6 | 53 | ~45 | 68 | ~101 |
| Neural spine length:neural arch length | 0.43* | 0.77 | 0.33 | 0.67 | 0.54 |

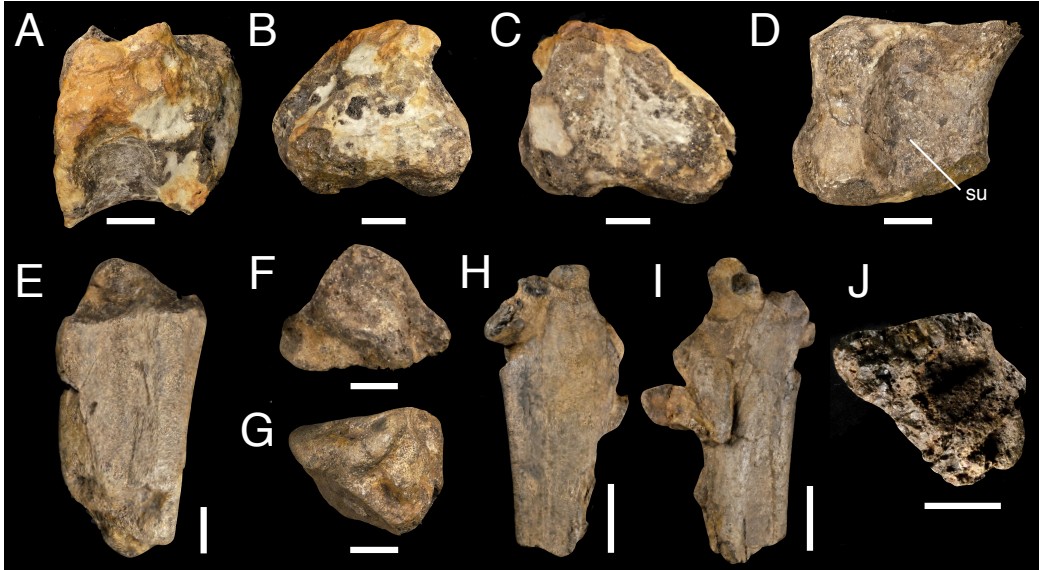

**Figure 6** **Sacrocaudal fragment IWCMS 2018.30.4 (A–D) and rib fragments IWCMS 2018.30.5 (E–G) and 2018.30.6 (H–J).** IWCMS 2018. 30.4 in (A) dorsal, (B) posterior, (C) anterior, (D) ventral views. IWCMS 2018.30.5 (E–G) and 2018.30.6 (H–J), views uncertain. Abbreviations: su, sulcus. Scale bars: 20 mm (A–G, J); 50 mm (H–I).

The longitudinal ventral sulcus of IWCMS 2018.30.4 suggests that this fragment might be an incomplete caudal centrum. Ventral sulci are common on theropod caudal vertebrae including those of spinosaurids (*Samathi, Sander & Chanthasit, 2021*), although we note that *Rauhut (2003)* did not observe any in the cf. *Suchomimus* caudal element MNN GDF 510. Whilst ventral sulci can be narrow in theropod caudal centra (*Rauhut (2003)*, they are broad in some taxa, including some large megalosaurids (*Rauhut et al., 2018*). Additionally, the fragment is similar in ventral view to the anterior caudal vertebrae of *Vallibonavenatrix* (*Malafaia et al. (2020b)*: Fig. 6E) which also possess a broad ventral sulcus bordered by parallel crests. The somewhat bevelled ventral portion of the posterior surface may be a chevron facet. However, we cannot exclude the possibility that IWCMS 2018.30.4 is a sacral vertebra: it is similar to the other sacral elements in width, and the presence of a ventral sulcus is a feature seen in spinosaurid sacral vertebrae, including those of *Vallibonavenatrix* (*Malafaia et al., 2020b*) and possibly *Camarillasaurus* (*Samathi, Sander & Chanthasit, 2021*).

*IWCMS 2018.30.5 and 6 (Rib fragments).* A pair of rib shaft fragments are preserved (Figs. 6E–6J), although it cannot be determined whether they pertain to the same element. The larger one, which is associated with a confused mess of bone fragments cemented to its surfaces, has a length of 194.0 mm. The other measures 144.3 mm and is largely well preserved despite the loss of its dorsal and ventral segments. A triangular cross-section with rounded corners is apparent in the latter, the widest of the three surfaces measuring 80.8 mm. Whilst this morphology was likely present ventrally in the larger piece (despite

the damage sustained to one of the margins), this fragment appears to flare and flatten dorsally. The internal cross-section of the smaller fragment is infilled with cancellous bone. Such internal organisation could not be reliably ascertained *via* macroscopic examination of the larger fragment's extremities.

### Appendicular elements

*IWCMS 2018.30.7 and 8 (Ilium fragments).* A pair of fragments representing a single, postacetabular process of a right-sided ilium were recovered. The fragments are poorly preserved and do not fit back together, though it would appear that only a slither of material is missing (Fig. 7). The fragments are large and robustly built, and lack any evidence of pneumaticity.

The remains of the brevis fossa can be distinguished, preserved as at least two separate pieces; the anterior piece measures ~135 mm (anteroposterior length), and the more posterior fragment ~145 mm. The medial side has been mostly stripped of its overlying cortical bone. The dorsally projecting postacetabular blade is missing, and what remains are medial and lateral blades that together enclose the brevis fossa. The former is incomplete and its extent difficult to assess, although it likely faced mainly ventrally. Enough of the ventrolaterally projecting lateral blade is well preserved to describe its generally thick and rounded morphology, posteriorly increasing ventrolateral projection, and flattened lateral surface. While stout anteriorly (with a dorsoventral thickness of 41.9 mm), it appears to thin posteriorly (dorsoventral height: 21.9 mm) before thickening again (dorsoventral height: 34.1 mm). When viewed ventrally, both pieces describe a posteriorly expanding fossa. A small neurovascular foramen is present on the anterior margin of the more anterior piece.

Additional fragments probably pertain to the ilium given their triradiate and triangular cross-section, but are very poorly preserved. These are briefly reported in the supplementary information.

*IWCMS 2018.30.9 and 10 (Long bone fragments).* Two transverse slices of a long bone are preserved (Fig. 8), one with a largely sub-circular cross-section while the other likely possessed a more ovate cross-section in life. Both are damaged and offer little of note bar their diameter (107.8 mm and 123.7 mm respectively) and asymmetrical cortical bone thickness. The space enclosed by the cortical bone is occupied by cancellous bone with no evidence of a medullary cavity, perhaps suggesting the pieces derived from the metaphyseal region of the limb bone. It is uncertain as to whether both belong to the same element, and to which element that may be, although we presume it originates from the pelvic limb given the rest of the material recovered for this individual.

### Theropod affinity of the material

Multiple lines of evidence suggest the material pertains to a large theropod dinosaur. Whilst the neural arch fossae and delimiting laminae support the saurischian affinities of IWCMS 2018.30.3 more generally (*Wilson et al., 2011*), the presence of a pneumatic foramen posterior to the parapophysis supports theropodan or neotheropodan affinities of the anterior presacral vertebra IWCMS 2018.30.1 (*Carrano, Benson & Sampson, 2012*;

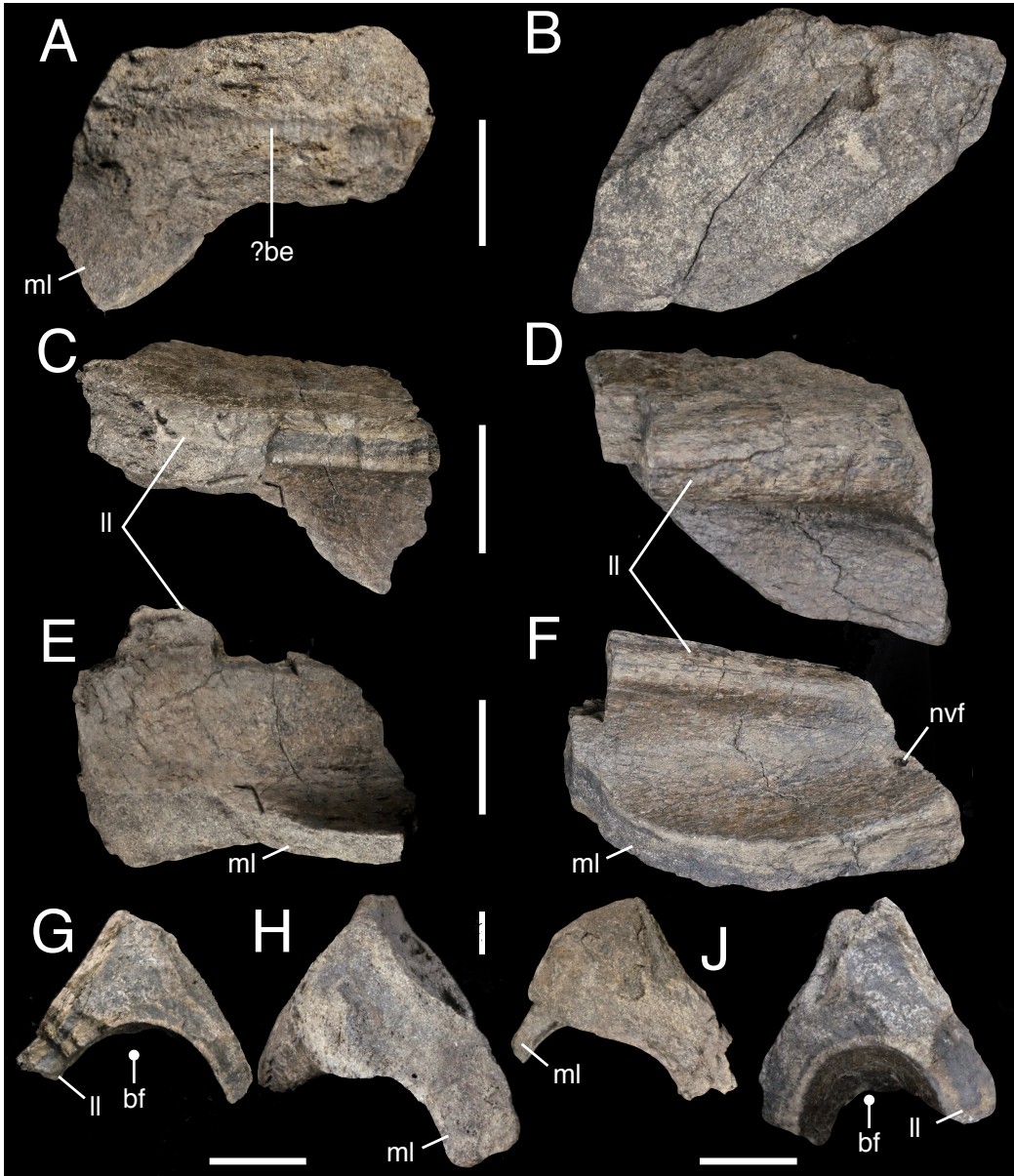

**Figure 7** **Fragmentary postacetabular process of the right ilium IWCMS 2018.30.7 (A, C, E, G, I) and 2018.30.8 (B, D, F, H, J).** In (A–B) medial, (C–D) ventrolateral oblique, (E–F) ventral, (G–H) anterior and (I–J) posterior views. Abbreviations: be, bioerosion; bf, brevis fossa; ll, lateral lamina; ml, medial lamina; nvf, neurovascular foramen. Scale bars: 50 mm.

*Cau, 2018*). The opisthocoelous condition of the latter's centrum (*Holtz, Molnar & Currie, 2004*) is common within the cervical and anterior dorsal vertebrae of non-coelurosaurian tetanurans; indeed, opisthocoely is synapomorphic of carnosaur cervicals in certain analyses (*Rauhut, 2003*; *Rauhut & Pol, 2019*) and is notably pronounced in allosauroids and megalosauroids (*Evers et al., 2015*). Elsewhere, the pronounced, well-developed brevis fossa of the ilium has been considered diagnostic of Theropoda in some previous works

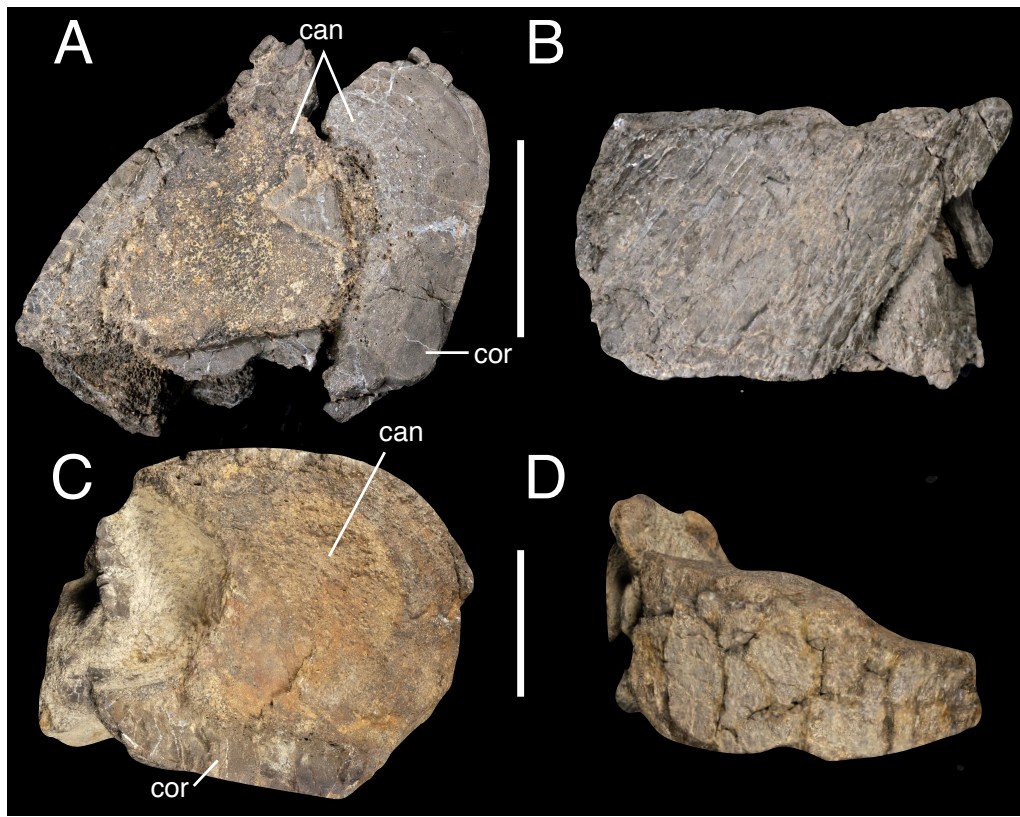

**Figure 8** **Long bone fragments IWCMS 2018.30.9 (A, B) and 2018.30.10 (C, D).** Views uncertain. Abbreviations: can, cancellous bone; cor, cortical bone. Scale bars: 50 mm.

(*Gauthier, 1986*), although a large and expanded brevis fossa on the ilium is observed for dinosaurs more generally (*Hutchinson, 2001*). Also of note is the relatively thin-walled nature of the long bones fragments, a trait also deemed synapomorphic for Theropoda (*Gauthier, 1986*).

Sauropods share opisthocoelous and pneumatic cervical and anterior dorsal vertebrae with some theropods (*Upchurch, Barrett & Dodson, 2004*; *Upchurch, Mannion & Barrett, 2011*) but several lines of evidence are inconsistent with a sauropod identity for the Compton Chine material. If a cervical position is assumed for IWCMS 2018.30.1 (see "Descriptive osteology" for further comments regarding element position), subdivision of the pneumatic foramen would be expected (*Upchurch, 1995*; *Whitlock, 2011*). Moreover, cervical ventral keels are rare in sauropods and their parapophyses—which are typically indented—consistently maintain a ventral position throughout the series (*Upchurch, Barrett & Dodson, 2004*). Similarly, if an anterior dorsal position is assumed, the element's generally abbreviated dimensions are inconsistent with a sauropod identity, since these vertebrae are the longest of the dorsal series in Sauropoda (*Upchurch, Barrett & Dodson, 2004*). In addition, while opisthocoelous and ventrally keeled cervical and anterior dorsal vertebrae are present in large ornithopod vertebrae from the Wealden Supergroup (*Norman, 2011*), skeletal pneumaticity is absent within Ornithischia (*Rauhut, 2003*). Further, the

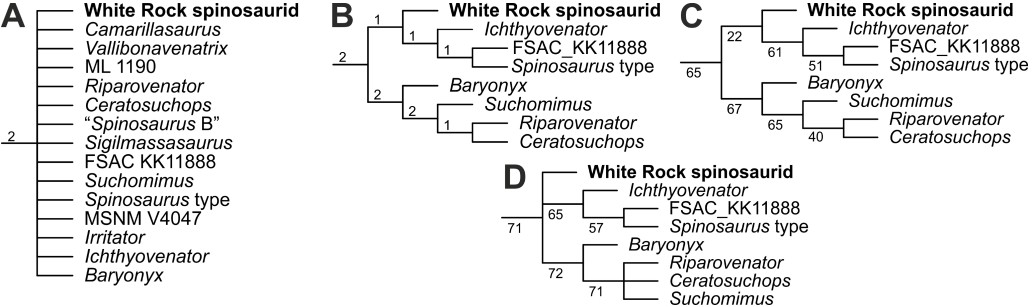

**Figure 9** Phylogenetic results following the addition of the White Rock spinosaurid to the modified dataset of *Barker et al. (2021)*, focusing on Spinosauridae. (A) Strict consensus tree; (B) reduced consensus tree showing stable spinosaurid OTUs; Jackknife values based on (C) absolute and (D) GC frequencies after wildcard OTUs were pruned. Numbers above and below nodes indicate Bremer and jackknife values respectively. Full versions available in the Supplemental Information.

proposed caudal element IWCMS 2018.30.3 lacks the ossified tendons present on the neural spines of ornithopod vertebrae near the pelvis (*Norman, 2011*), and lacks the rectangular outline of the anterior caudal vertebrae of basal iguanodontians (*Norman, 2004*). Referral to either Sauropoda or Ornithopoda can thus be rejected.

More specifically, the flattened peripheral rim around the anterior articular surface observed in IWCMS 2018.30.1 is characteristic of megalosaurian cervical vertebrae (*Carrano, Benson & Sampson, 2012*), although it can be observed in anterior dorsal vertebrae as well (*e.g.*, *Baryonyx*; *Charig & Milner, 1997*). Additionally, the presence of spinodiapophyseal webbing in IWCMS 2018.30.3 is characteristic of spinosaurid dorsal vertebrae (or various spinosaurid in-groups, depending on the analysis) (*Barker et al., 2021*; *Benson, 2010*; *Carrano, Benson & Sampson, 2012*; *Evers et al., 2015*; *Holtz, Molnar & Currie, 2004*; *Rauhut, 2003*; *Rauhut & Pol, 2019*) and have been documented in spinosaurid anterior caudal vertebrae as well (*Barker et al., 2021*; *Samathi, Sander & Chanthasit, 2021*). *Coria & Currie (2016)* described the presence of webbing in the dorsals of some megaraptorans, although the clade currently lacks any presence in the European record (*White et al., 2020*). Thus, combined with our phylogenetic results (see "Phylogenetic analysis"), we consider the presently discussed material to pertain to a large spinosaurid.

## Phylogenetic analysis

The New Technology Search returned 30 trees of 2,451 steps and consistency (excluding the 1,068 parsimony uninformative characters deactivated using the command *xinact*), rescaled consistency, and retention indices (CI, RCI and RI) of 0.374, 0.171 and 0.456 respectively. The round of TBR recovered 22,535 trees. The strict consensus tree finds Spinosauridae to be completely unresolved (Fig. 9A). Seven other spinosaurid OTUs (*Irritator*, MSNM V4047, *Sigilmassasaurus*, '*Spinosaurus* B', ML 1190, *Vallibonaventrix* and *Camarillasaurus*) were identified as wildcard taxa following the iterPCR method.

Interestingly, the reduced consensus recovered a baryonychine-spinosaurine split, with the White Rock spinosaurid placed as an early-branching member of Spinosaurinae

(Fig. 9B), albeit with limited support. Three characters were shared between the White Rock spinosaurid and other spinosaurines, all from the anterior caudal series: the presence of centrodiapophyseal laminae (Ch. 358:1), the presence of prezygodiapophyseal laminae (Ch. 626:1), and the presence of a deep prezygocentrodiapophyseal fossa (Ch. 1605:1). Jackknife resampling (Figs. 9C, 9D) also recovered low nodal support (both absolute and GC frequency values), with the White Rock spinosaurid instead assuming a position amongst Spinosaurinae or recovered in a polytomy outside Baryonychinae and Spinosaurinae.

## REMARKS

### The White Rock spinosaurid: a British spinosaurine?

The recovery of the White Rock spinosaurid as an early branching member of Spinosaurinae in some data runs is intriguing, especially considering the current absence of the clade from Lower Cretaceous deposits of the British Isles. Spinosaurines may have originated in Europe (*Barker et al., 2021*), and phylogenetic and quantitative analyses of fragmentary materials support their presence in the quasi-contemporaneous deposits of Iberia (*Alonso & Canudo, 2016*; *Alonso et al., 2018*; *Isasmendi et al., 2020*; *Malafaia et al., 2020a*; *Sánchez-Hernández, Benton & Naish, 2007*). The three above-listed spinosaurine synapomorphies were also recovered in the previous iteration of the analysis used here (*Barker et al., 2021*).

However, the distribution of these three caudal character states could potentially be a function of the relative position of these elements along the axial column. Indeed, specimens such as FSAC-KK 11888 (*Ibrahim et al., 2020a*) and MN 4743-V (*Bittencourt & Kellner, 2004*) appear to show that fossae and laminae become less prominent in the more posterior parts of the axial skeleton. We consider IWCMS 2018.30.3 to be more anteriorly placed than any of the known caudal elements of *Riparovenator* or *Vallibonavenatrix* (specimens that are also known from anterior caudal material); scores regarding fossae or laminae for the latter pair's anterior caudal series might thus be affected by a lack of positional overlap. Comparisons are exacerbated by our incomplete knowledge of the anteriormost caudal series of other relevant taxa, such as *Baryonyx* and *Suchomimus* (*Charig & Milner, 1997*; *Sereno et al., 1998*). In addition, the presence of centrodiapophyseal (Ch. 358:1) and prezygodiapophyseal laminae (Ch. 626:1) is not unique to Spinosaurinae: rather, these character states are homoplastic amongst tetanurans. Our understanding of character distribution within spinosaurid tails would very obviously benefit from the discovery of more complete (*i.e.,* overlapping) anterior caudal vertebrae from non-spinosaurine taxa.

In sum, we do not consider the recovered synapomorphies to be sufficiently diagnostic or the nodal support sufficiently robust to warrant referral of the White Rock spinosaurid to Spinosaurinae at this time.

### Further comparisons

The presence of a sub-parapophyseal sulcus in the probable dorsal vertebra IWCMS 2018.30.1 is similar to the (albeit better developed) sulci described in the anterior dorsal centrum of the indeterminate tetanuran *Vectaerovenator* (*Barker et al., 2020*). Similarly positioned sulci are present in the possible megalosauroid *Yunyangosaurus* (*Dai et al., 2020*). While *Vectaerovenator*'s incomplete nature requires that its phylogenetic position

remains ambiguous, it is interesting that constrained phylogenetic analyses found that few extra steps were required to recover it within Megalosauroidea (*Barker et al., 2020*) and it possesses at least some features (including enlarged pneumatic foramina) akin to the synapomorphic condition of megalosaurian anterior dorsal centra (*Carrano, Benson & Sampson, 2012*). However, caution is advised when discussing this character in IWCMS 2018.30.1, given the state of preservation on the contralateral side that precludes assessment of any mirroring.

The possible presence of a median tuberosity in IWCMS 2018.30.1 is similar to that observed in the posterior cervical and anterior dorsals of *Sigilmassasaurus* (*Evers et al., 2015*), and would suggest the feature is more broadly distributed amongst spinosaurids. What remains of the ventral keel in this specimen is prominent and straight, as seen in Spinosauridae (*Barker et al., 2021*; *Evers et al., 2015*). The robust ventral keel differs from theropods more generally, however, with anterior dorsal centra in particular typically producing deep, sharp keels (*Rauhut, 2003*). However, robust keels may occur around the cervicodorsal region and are perhaps a function of overall size, given the tendency for increased keel robusticity in larger elements of some spinosaurid material (*Evers et al., 2015*).

The shallowly concave, nearly horizontal lateral profile of the ventral margins of the sacral vertebrae (IWCMS 2018.30.2) is typical of many theropods. They lack the strongly arched condition of various ceratosaurs (*Carrano, Benson & Sampson, 2012*; *Carrano & Sampson, 2008*; *Rauhut & Pol, 2019*). The anteroposteriorly elongate centra are similar to those of other spinosaurids including *Suchomimus*, *Vallibonavenatrix* and *Camarillasaurus*, although such dimensions also occur in some ceratosaurs and *Megalosaurus* (*Samathi, Sander & Chanthasit, 2021*). The presence of a ventral sulcus on the posterior sacral centrum recalls a similar structure on the third sacral of *Vallibonaventrix* (*Malafaia et al., 2020b*) but it does not extend as far anteriorly in the White Rock spinosaurid. The sacral centra also recalls *Vallibonaventrix* and the lost *Spinosaurus aegyptiacus* type specimen (*Stromer, 1915*) in possessing depressed lateral surfaces. So called sacral "pleurocentral depressions" have been deemed synapomorphic for Allosauria and Megalosauridae in some analyses (*Carrano, Benson & Sampson, 2012*; *Rauhut & Pol, 2019*), but are also present in various coelurosaurs (*Holtz, Molnar & Currie, 2004*), with those of IWCMS 2018.30.2 poorly developed compared to such taxa as *Megalosaurus* (*Benson, 2010*) and *Allosaurus* (*Gilmore, 1920*). As above, we consider the features in IWCMS 2018.30.2 to represent non-pneumatic lateral indentations; the centra thus remain apneumatic, as is typical of non-avian theropods but contrasts with the condition in *Vallibonavenatrix* (*Malafaia et al., 2020b*).

The anteroposteriorly narrow neural spine (relative to neural arch length) of IWCMS 2018.30.3 differs from longer condition observed in the "pelvic" axial series (*i.e.*, the vertebral series encompassing the posterior dorsals to the anterior caudals) of such spinosaurids as *Baryonyx* (*Charig & Milner, 1997*), *Ichthyovenator* (*Allain et al., 2012*) and *Suchomimus* (the latter only preserves large, sheet-like neural spine tips in its anterior caudal series; *Sereno et al. (1998)*: Fig. 3). When caudal elements are compared (Table 4), IWCMS 2018.30.3 is closest to *Riparovenator*, although (as mentioned previously) we

consider the anteriormost preserved caudal element of the latter to occupy a comparatively more posterior position. Indeed, IWCMS 2018.30.3 differs from *Riparovenator* in the absence of an anterior spur (=accessory neural spine of some) at the base of the neural spine. Anterior spurs are more common towards the mid-caudal series in taxa possessing this feature (*Rauhut, 2003*), and are similarly absent from the anteriormost elements of *Ichthyovenator* (*Allain et al., 2012*) and the entirety of the caudal series of FSAC-KK 11888 (*Ibrahim et al., 2020a*).

Additionally, the lack of postzygocentrodiapophyseal fossae in IWCMS 2018.30.3 suggests a difference in centrodiapophyseal fossae morphology in this individual relative to some other spinosaurids. Three centrodiapophyseal fossae are present in the neural arches of the anterior caudal vertebrae of such specimens as the spinosaurine FSAC-KK 11888 (*Ibrahim et al., 2020a*), MN 4743-V (*Bittencourt & Kellner, 2004*), and the 'Phuwiang spinosaurid B' material (SMPW9B-14, 15) (*Samathi, Sander & Chanthasit, 2021*). However, as noted above, more convincing comparison can only take place when better corroboration pertaining to the proposed axial position of IWCMS 2018.30.3 occurs. Elsewhere on IWCMS 2018.30.3, the presence of pleurocentral depressions is also shared with the anterior caudal vertebrae of *Vallibonavenatrix* (*Malafaia et al., 2020b*) and *Iberospinus* (*Mateus & Estraviz-López, 2022*), as well as the megalosaurids *Torvosaurus*, *Megalosaurus* and *Wiehenvenator* (*Rauhut, Huebner & Lanser, 2021*; *Rauhut & Pol, 2019*).

The posteriorly diverging margins of the brevis fossa (IWCMS 2018.30.7, 8) recall the condition in *Baryonyx* (*Charig & Milner, 1997*) and *Vallibonaventarix* (*Malafaia et al., 2020b*); indeed, this character state has previously been suggested as a synapomorphy of Baryonychinae *sensu Barker et al. (2021)*. It is, however, also wide in the spinosaurine FSAC KK11888 (O. Rauhut, pers. comm., 2022). In *Ichthyovenator*, a taxon recovered in *Barker et al. (2021)* as a spinosaurine but whose affinities are not entirely clear (*Evers et al., 2015*), the fossa is narrow and with subparallel margins (*Allain et al., 2012*). Posterior expansion of the brevis fossa is nevertheless common in Neotheropoda (*Carrano, Benson & Sampson, 2012*) and is present in a variety of tetanurans (*Benson, 2010*), indicating a wider distribution of the character state.

## Brief biostratinomic comments

All elements that make up the specimens described here are highly fragmented. The transverse slices of long bone show variation in cortical thickness, perhaps exacerbated by varying degrees of delamination. Other elements display cracked, crazed and irregular surface markings. The best-preserved bones—the fused sacral vertebral centra (Fig. 4)—show longitudinal cracking, while some other bored elements (see below; Fig. 10) possess reasonably preserved cortex on one surface but roughened, irregular looking cortical surfaces elsewhere. These changes equate to stages 1–3 in *Behrensmeyer's (1978)* scale of weathering and abrasion, suggesting a possible pre-burial interval of 3–4 years. Given the highly fragmentary state, we note that trampling may also have occurred (*Britt et al., 2009*), and perhaps accounts for the crushed in left lateral surface of IWCMS 2018.30.3 in particular.

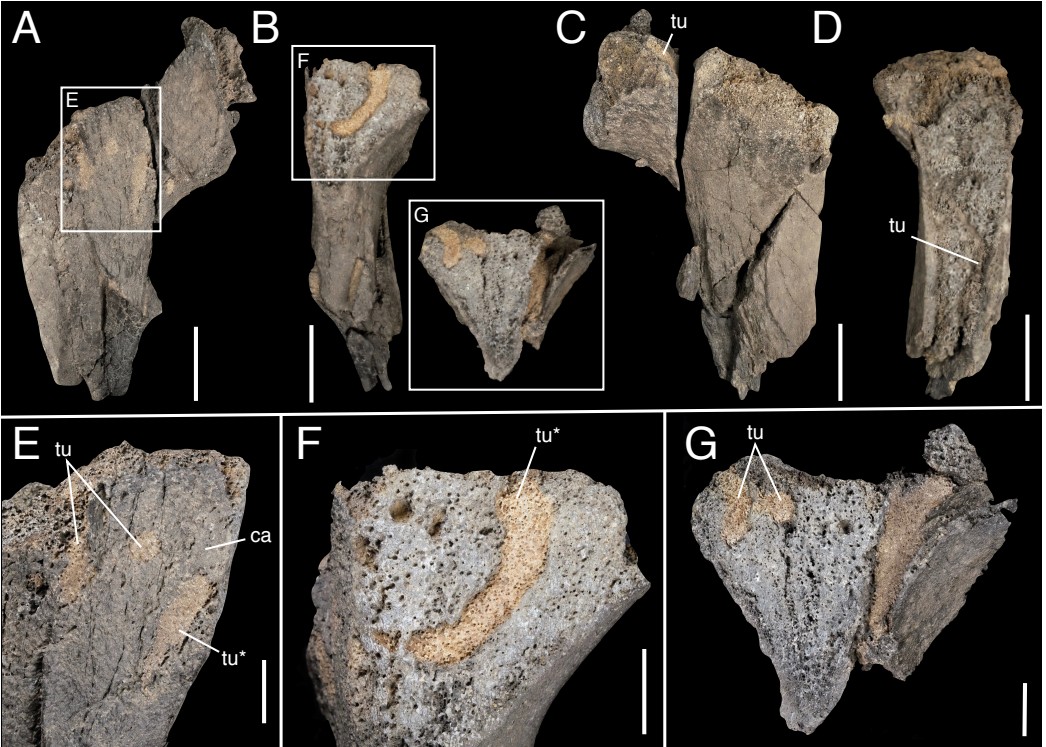

**Figure 10** **Bioeroded indeterminate bone fragment IWCMS 2018.30, displaying cross-sections of internal tubes.** Views uncertain. (F) and (G) are counterparts. Asterisks denote continuation of a single tube visible in different views. Abbreviations: ca, cancellous bone; tu, tubes (preserved in cross-section). Scale bars: 50 mm (A–D); 20 mm (E–G).

Bioerosion, represented by curved tubes of uniform width, is present on several elements and is interpreted as representing invertebrate feeding traces (Figs. 10A–10G). These extend into the cancellous bone for ∼80 mm and have circular cross-sections with a diameter of ∼10 mm. Terrestrial bone borings with equivalent diameters have been recorded in the Upper Jurassic and throughout the Cretaceous (*Britt, Scheetz & Dangerfield, 2008*; *Csiki, 2006*; *Paik, 2000*; *Rogers, 1992*). In all cases, beetles (Coleoptera) were considered the most likely bioeroders. No bioglyphs are visible on our specimen, although the boring infills have been left in situ. When reassembled, the more medially placed circular cross-section in Fig. 10G abuts the marginally placed end of the longitudinal section of its counterpart in Fig. 10F, indicating the possibility of a right-angled branch or direction change. The borings were infilled by matrix and macroscopic bone chippings or frass are absent. This suggests that burial occurred after the bioerosion occurred.

*Britt, Scheetz & Dangerfield (2008)* considered borings more than five mm in depth to be ethologically indicative of internal mining or harvesting of bone. Necrophagous coleopterans and their larvae (in particular desmestids) are among the most common invertebrate bone modifiers (*Xing et al., 2013*) and feed on desiccated carcasses that are subaerially exposed (*Bader, Hasiotis & Martin, 2009*; *Cruzado-Caballero et al., 2021*); osteophagy occurs when other food sources are exhausted (*Bader, Hasiotis & Martin,*

*2009*), bone borings being more typically related to pupation (*Höpner & Bertling, 2017*). Regardless, bioerosion created by dermestid-type beetles can involve the creation of tunnel (=tube)-like structures (*Britt, Scheetz & Dangerfield, 2008*; *Cruzado-Caballero et al., 2021*; *Höpner & Bertling, 2017*).

Circumstantial support for the possible importance of dermestids as bone modifiers in Wealden environments is provided by the existence of this group in the Middle Jurassic (*Deng et al., 2017*) and the fact that beetles are the most abundant Wealden Supergroup insect, the caveat here being that they are mostly represented by elytra (which are largely non-diagnostic to family level; *Jarzembowski, 2011*).

Several other necrophagous insect groups can be excluded from consideration (*Bader, Hasiotis & Martin, 2009*; *Cruzado-Caballero et al., 2021*; *Xing et al., 2013*): hymenopterans and isopterans typically produce star-shaped features and isopterans tend to cause more widespread, irregular damage, rather than tunnels (*Hutchet, 2014*); tineid moths (Lepidoptera) specialise in keratinous tissues and traces made by them have yet to be identified in the fossil record; and the burrows of mayfly (Ephemeroptera) larvae are typically narrow, U-shaped, thin walled, and limited to aquatic environments. Damage by other aquatic organisms such as burrowing bivalves are also improbable given the taphonomic circumstances and the curved form of the structures (such molluscs usually produce clavate-shaped borings; *McHugh et al., 2020*), whilst the parallel-sided morphology with lack of splitting makes plant root damage unlikely (*Rogers, 1992*).

An additional trace can be observed on the abraded medial surface of a fragment of ilium. It takes the form of a straight, wide, parallel-sided 'furrow' that extends across the exposed cancellous bone (Fig. 7A) (at mid-length, some of the furrow's margins have seemingly been eroded). As furrows typically describe open excavations affecting cortical bone (*Britt, Scheetz & Dangerfield, 2008*; *Pirrone, Buatois & Bromley, 2014*), this structure might represent one side of a tube akin to those described above. Additional divot-like impressions are present on other pieces of the ilium, but these are difficult to separate from non-biological damage and are not considered further here. Elsewhere, several tooth mark-like traces are observed on the smaller rib fragment. However, they likely do not represent vertebrate feeding traces (D. Hone, pers. comm., 2021). In sum, we tentatively attribute the traces to coleopteran bioerosion related to harvesting behaviour, but note that additional study is required.

## DISCUSSION

The presence of multiple theropod—and specifically spinosaurid—characters across various elements, combined with the consistency in specimen size, preservation and adhering matrix, supports their referral to a single spinosaurid individual. Given the material's state of preservation, more precise identification is not currently possible, and the specimen is best classified as Spinosauridae indet. The White Rock spinosaurid likely does represent a new taxon, but we are unable to diagnose it based on the material to hand.

The discovery of this specimen in the basal unit of the Vectis Formation renders it the youngest documented spinosaurid material from the Wealden Supergroup. Previous finds

from the Wealden Group had been restricted to the underlying Wessex Formation (*Barker et al., 2021*; *Hutt & Newbery, 2004*; *Martill & Hutt, 1996*) and no spinosaurid material is known from equivalent outcrops in Dorset (*Penn et al., 2020*). A possible contemporary is perhaps represented be a worn tooth crown (NHMUK PV R 5165, initially referred to *Goniopholis crassidens*) recovered from Atherfield on the Isle of Wight (*Fowler, 2007*), a locality that contains outcrops of the Vectis Formation. Unfortunately, precise stratigraphic information is missing for this specimen.

Comparisons with the spinosaurid record from the younger members of the neighbouring Weald Clay Group are more difficult. The Upper Weald Clay Formation yielded the type specimen of *Baryonyx walkeri* (*Charig & Milner, 1986*) and is largely synchronous with the exposed Wealden Group strata on the Isle of Wight. The base of this formation is Barremian in age, but its upper age has proven difficult to constrain and may be late Barremian or early Aptian (*Radley & Allen, 2012b*); indeed, the palynomorph, ostracod and mollusc faunas of the upper units of the Upper Weald Clay Formation are similar to those of the Vectis Formation (*Radley & Allen, 2012b*). However, the *Baryonyx walkeri* type specimen was recovered from Smokejacks Pit in Ockley, Surrey, whose exposures in the Upper Weald Clay Formation are consistent with an early Barremian age (*Radley & Allen, 2012b*; *Ross & Cook, 1995*). A baryonychine tooth crown (MNEMG 1996.133) was recovered from Ewhurst's Brickworks (Surrey) from a layer equivalent to the top of the Smokejacks beds (*Charig & Milner, 1997*). We are unaware of any younger spinosaurid occurrences from the Weald Clay Group, although the historical nature of many accessioned Wealden specimens renders it difficult to collate precise stratigraphic information. Nevertheless, spinosaurids are known from the late Barremian and early Aptian of Iberia (*Malafaia et al., 2020a*), suggesting the potential existence of younger British specimens.

Despite the general rarity of Vectis Formation dinosaur remains, ichnological evidence from the White Rock Sandstone suggests the sandflat facies supported large dinosaur populations that visited the fluctuating, plant colonised shoreline (*Radley & Allen, 2012c*; *Radley, Barker & Harding, 1998*). More generally, the recovery of spinosaurid remains from this formation is perhaps expected. Not only are its units within the temporal span of the clade, spinosaurid remains from lagoonal deposits have been documented elsewhere (see *Bertin (2010)* for a review of depositional environments containing spinosaurid remains), and their occurrences have been shown to correlate with 'coastal' palaeoenvironments (relative to other sampled taxa) (*Sales et al., 2016*), a broad category that includes paralic environments (*Butler & Barrett, 2008*).

A remarkable feature of the White Rock spinosaurid is its large size (Table 5). Large theropods from the underlying Wessex Formation include the allosauroid *Neovenator salerii* (*Brusatte, Benson & Hutt, 2008*; *Hutt, Martill & Barker, 1996*) and the spinosaurids *Ceratosuchops* and *Riparovenator* (*Barker et al., 2021*). While ichnological evidence reinforces the presence of particularly large forms in the Wessex Formation (*Lockwood, 2016*), the size of the White Rock spinosaurid appears to eclipse that of the above taxa, as well as other European theropods.

Barker et al. (2022), *PeerJ*, DOI 10.7717/peerj.13543
**Table 5 Comparative dorsoventral heights (in millimetres) of the posterior articular facets of the caudal vertebrae of various tetanurans.** Where several caudal vertebrae are known, the largest is presented here. Note that only data for the anterior articular facet is available for the lost *Spinosaurus* holotype and FSAC KK-11888 (marked by an asterisk (*)). Data collected from *Stromer (1915)*; *Dong, Zhou & Zhang (1983)*; *Charig & Milner (1997)*; *Brochu (2003*: Fig. 59A); *Allain et al. (2012)*; *Hendrickx & Mateus (2014)*; *Rauhut et al. (2018)*; *Ibrahim et al. (2020a)*; *Samathi, Sander & Chanthasit (2021)* and *Mateus & Estraviz-López (2022)*. Measurements for *Riparovenator* taken by CTB.

| Spinosauridae | | | | | | | | Other tetanurans | | | |
|---|---|---|---|---|---|---|---|---|---|---|---|
| Spinosauridae indet. (IWCMS 2018.30.3) | *Baryonyx* (NHMUK PV R9951) | *Riparovenator* (IWCMS 2020.447.7) | *Ichthyovenator* (MDS BK10-03) | *Spinosaurus* (BSP 1912 VII 19) | Spinosaurinae indet. (FSAC KK-11888) | Phuwiang spinosaurid B" (SM-PW-9B-17) | *Iberospinus* (ML 1190-15) | *Tyrannosaurus rex* (FMNH PR2081) | *Yangchuanosaurus magnus"* (CV00216) | *Torvosaurus gurneyi* (ML 1100) | Megalosaurinae indet. (MUJA-1913) |
| 159.8 | 110 | 125.6 | 128 | 135* | 129* | 88 | 100.7 | ~257 | 140 | 145 | 150 |

The fragmentary megalosaurine caudal vertebra MUJA-1913 is currently regarded as the largest European theropod skeletal material (based on the dorsoventral height of its posterior articular facet). Its size suggests an individual more than 10m in length (*Rauhut et al., 2018*). A set of large caudal vertebrae from the Oxfordian (Jurassic) of France with potential megalosaurid affinities are said to be of comparable size, but have yet to be published in detail (*Pharisat, 1993*; *Rauhut et al., 2018*). IWCMS 2018.30.3 exceeds the dorsoventral proportions of MUJA-1913 (Table 5). Similarly, the anterior sacral vertebra of the White Rock spinosaurid is larger anteroposteriorly (∼156 mm) than that of spinosaurids for which data is known, including *Vallibonavenatrix* (five recovered vertebrae, length range: 90–96 mm) (*Malafaia et al., 2020b*) and FSAC KK-11888 (three vertebrae, length range: 135–145 mm) (*Ibrahim et al., 2014*), being sub-equal to the largest sacral element of the *Spinosaurus* type specimen (of the three recovered vertebrae, lengths for the two most complete ones are >130 mm and 155 mm) (*Stromer, 1915*). The brevis fossa in IWCMS 2018.30.7 also supports these extrapolations: the maximum measurable width is 84.6 mm but the fossa probably flared to a greater width when complete. In comparison, the fossa has a maximum width of ∼50 mm in *Ichthyovenator* (based on (*Allain et al., 2012*: Fig. S7), 60 mm in *Vallibonavenatrix* (*Malafaia et al., 2020b*), and ∼70 mm in *Allosaurus* (based on *Madsen, 1976*, pl. 46B).

*Aureliano et al. (2018)* suggested that the evolution of large body sizes (*i.e.,* 10–15 m) in Spinosaurinae may be linked to their semi-aquatic specialisations; indeed, selection for increased size has been noted amongst aquatic vertebrates in general (*Gearty, McClain & Payne, 2018*; *Heim et al., 2015*). However, the definition of 'semi-aquatic' remains problematic within the context of spinosaurid ecology; not only is the degree of aquatic adaptation within spinosaurines a disputed issue (*Hone & Holtz Jr, 2019*), there is also the fact that the apparently less aquatic baryonychines (*Arden et al., 2019*; *Hone & Holtz Jr, 2021*), such as *Suchomimus*, also exceeded 10 m (*Sereno et al., 1998*; *Therrien & Henderson, 2007*). At the time of writing the degree and nature of aquatic adaptations within spinosaurids remains the topic of research (*Barker et al., 2017*; *Fabbri et al., 2022*; *Henderson, 2018*; *Hone & Holtz Jr, 2019*; *Hone & Holtz Jr, 2021*; *Ibrahim et al., 2020a*; *Ibrahim et al., 2014*); nevertheless, it is not clear that giant size in Spinosaurinae is linked to aquatic habits. Indeed, the especially large baryonychine *Suchomimus* was recently inferred to be less aquatically adapted than the "subaqueous" foragers *Baryonyx* or material referred to *Spinosaurus* (*Fabbri et al., 2022*) on the basis of histological data. If valid, this indicates a lack of correlation between size and aquatic ecology. Moreover, these histological results are not incompatible with the wading hypothesis suggested for Spinosauridae (*Hone & Holtz Jr, 2021*), rather than the more specialised "subaqueous foraging" ecology suggested for *Baryonyx* and cf. *Spinosaurus* in particular (*Fabbri et al., 2022*). The discovery of the large-bodied White Rock spinosaurid, lacking unambiguous spinosaurine affinities or obvious traits suggestive of enhanced aquatic specialisation (*e.g.,* the long bone cross-sections do not appear to be particularly dense), also lends support to this contention. Histological sectioning of this material, and comparison to results collected for other spinosaurids (*Fabbri et al., 2022*), would nevertheless be beneficial, especially given the

limited histological data known for spinosaurids (*Cullen et al., 2020*). Such analysis is beyond the scope of the present report.

In sum, whilst the precariousness of extrapolating overall body size from singular bones and dimensions cannot be understated, the impressive proportions of the White Rock spinosaurid material (IWCMS 2018.30.3 in particular) demonstrate the presence of a notably large tetanuran in the Wealden Supergroup of Britain: one that rivalled or even exceeded the largest theropods recovered elsewhere from the European Mesozoic.

## CONCLUSIONS

The White Rock spinosaurid represents the first documented spinosaurid from the Vectis Formation of the Isle of Wight, extending the temporal span of the clade in the British fossil record to the late Barremian. This stratigraphic positioning also renders it the youngest spinosaurid known from the UK. The White Rock spinosaurid is likely a novel taxon: however, the specimen lacks convincing autapomorphies and we presently opt to identify the specimen as Spinosauridae indet. Our phylogenetic analysis was unable to resolve its position within Spinosauridae but weakly supported spinosaurine or early-branching spinosaurid affinities were recovered in some data runs. Though fragmentary, it is the largest theropod currently known from the Wealden Supergroup, with some metrics exceeding those of the largest theropods known from Europe more generally.

**Institutional Abbreviations**

| | |
|---|---|
| **FSAC** | Faculté des Sciences, Casablanca University, Casablanca, Morocco |
| **IWCMS** | Isle of Wight County Museum Services, Dinosaur Isle Museum, Isle of Wight, UK |
| **MDS** | Dinosaur Museum, Savannahket, Laos |
| **MN** | Museu Nacional, Rio de Janeiro, Brazil |
| **MNBN** | Musée National Boubou Hama, Niamey, Niger |
| **MNEMG** | Maidstone Museum, Kent, UK |
| **MNN** | Musée National du Niger, Niamey, Niger |
| **MSM** | Museo Paleontológico Juan Cano Forner, Sant Mateu, Castellón, Spain |
| **MUJA** | Museo del Jurásico de Asturias, Colunga, Spain |
| **NHMUK** | Natural History Museum, London, UK |
| **SM** | Sirindhorn Museum, Department of Mineral Resources, Kalasin, Thailand |

## ACKNOWLEDGEMENTS

We thank Serjoscha Evers and Steve Hutt for useful comments regarding spinosaurid vertebral anatomy, Dave Hone for thoughts on vertebrate feeding traces, Steve Vidovich for advice on the phylogenetic analysis, Phil James (ilium and long bone fragments) and Mick Green (axial elements) for their expert preparation of the material, and Mark Penn for kindly donating a vertebral fragment. We also thank editor Andrew Farke and the two reviewers, whose comments and expertise improved the manuscript. The program TNT is made available thanks to the Willi Hennig Society.

### Funding

This work was supported by the EPSRC and the Institute for Life Sciences (IfLS), University of Southampton. The funders had no role in study design, data collection and analysis, decision to publish, or preparation of the manuscript.

### Grant Disclosures

The following grant information was disclosed by the authors:
The EPSRC and the Institute for Life Sciences (IfLS), University of Southampton.

### Competing Interests

The authors declare there are no competing interests.

### Author Contributions

- Chris T. Barker conceived and designed the experiments, performed the experiments, analyzed the data, prepared figures and/or tables, authored or reviewed drafts of the article, and approved the final draft.
- Jeremy A.F. Lockwood conceived and designed the experiments, analyzed the data, authored or reviewed drafts of the article, and approved the final draft.
- Darren Naish analyzed the data, authored or reviewed drafts of the article, and approved the final draft.
- Sophie Brown performed the experiments, authored or reviewed drafts of the article, and approved the final draft.
- Amy Hart performed the experiments, authored or reviewed drafts of the article, and approved the final draft.
- Ethan Tulloch performed the experiments, authored or reviewed drafts of the article, and approved the final draft.
- Neil J. Gostling conceived and designed the experiments, analyzed the data, authored or reviewed drafts of the article, and approved the final draft.

### Data Availability

The data matrix used in the phylogenetic analysis is available in the Supplementary Files.

### Supplemental Information

Supplemental information for this article can be found online at http://dx.doi.org/10.7717/peerj.13543#supplemental-information.

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
