# Peer review of "A European giant: a large spinosaurid (Dinosauria: Theropoda) from the Vectis Formation (Wealden Group, Early Cretaceous), UK"

_PeerJ, doi:10.7717/peerj.13543_

## Round 0.1 · original submission · Minor Revisions

This manuscript presents an intriguing new occurrence of a probable spinosaurid from the Vectis Formation. The descriptions are comprehensive, figures are very well done, and the interpretations are appropriately conservative given the nature of the fossils.

The reviewers outline a few relatively minor suggestions; I highlight two in particular:

1) Both reviewers request some additional text on the presumed association of the fossils described here, especially for IWCMS 2018.30.4. Please address this in revision.

2) The Systematic Paleontology section should go before the descriptions, to align with convention for most other papers; I do think it is OK to keep the section in the paper, even if a new taxon is not described.

·

Basic reporting

no special comments - see below for general comments

Experimental design

no special comments - see below for general comments

Validity of the findings

no special comments - see below for general comments

Additional comments

I found the paper of interest and a useful addition to our knowledge of European theropods, and have generally few comments. Most of my comments and corrections have been added directly on the annotated manuscript file, which is attached.
A few points of note:

I think the structure of the manuscript would need some improvements. It would be useful to have the descriptions of the material prior to the comments on its theropod and spinosaurid affinitities and the results of the phylogenetic analysis, as knowledge of the morphology is needed to understand these sections. As no new taxon is described, I also don't think that a formal "Systematic Palaeontology" section is needed; however, if you opt to keep it, please place it before the descriptions. Likewise, the comments on the preservation of the remains seems somewhat out of place; I would suggest to move this section to follow the geological setting or the overview of the materials.

Whereas I agree that the anterior dorsal, sacral and caudal vertebrae most probably belong to the same individual, I am skeptical about IWCMS 2018.30.4. This seems much to small to represent the same individual, and what is preserved of the vertebral centrum rather reminds me of ornithopod caudals.

In the phylogenetic analysis, some aspects remain somewhat unclear. First, you state that the analysis had 41 OTUs scored for 1810 characters - are all of these characters parsimony informative for your 41 OTUs? Based on the fact that the shortest trees have only 2451 steps, I think that this is highly unlikely. If you have a high amount of uninformative characters, mentioning of consistency indices does not make much sense, as these will be boostered by uninformative characters. You also mention that these are 1810 binary characters - are there really no multistate characters among such a high number of characters? Although you applied the iterPCR method, it also remains unclear if you computed a reduced conse´nsus tree and what the topology of that is - only an agreement subtree and the identification of problematic taxa is mentioned. Finally, I am somewhat uncertain why you used jackknifing results to state that spinosaurine relationships are not found under this option. As with bootstrapping or Bremer support, jackknifing is simply a method to evaluate robustness of results, but they do not represent an independent test of phylogenetic relationships. Given the extremely fragmentary nature of this specimen (and many other megalosauroids and spinosaurids in general, for that matter), very low robustness of the results is expected.

Reviewer 2 ·

Basic reporting

The text is clear, and the work is well-structured.
The figures are relevant, have good quality, are correctly labeled, and well described.
The conclusions are well stated and supported by the results of the performed analysis.
The literature references are adequate, and an update background was provided.

Experimental design

The authors describe an interesting set of osteological remains that increases the fossil record of dinosaurs known in the upper Barremian-lower Apian levels of the Wealden Group. These materials are interpreted as belonging to a giant spinosaurid, which would be the first evidence of this clade in the Vectis Formation and the youngest British record of these theropods.
The methods were described with sufficient detail so the analysis can be replicate.

Validity of the findings

The conclusions are well stated and limited to supporting results. However, there are two questions that, in my opinion, needs more detailed discussion:
(1) The authors interpreted all the remains as belonging to the same individual, but the materials were found on the foreshore and collected over a period of several months. The authors said that taphonomic and anatomical evidence show that they belong to a single individual, but do not explicit which data, beside possibly the large size and the putative taxonomic affinity, support this interpretation. I think that this question deserves some more detailed discussion in order to demonstrate that all the elements belong to the same individual.
(2) Please describe a little more detailed the preservation of the sacral centra. From the images it seems that both vertebrae preserve the articular surfaces, and it is curious the fusion of only two vertebrae in the sacrum of such a large-sized individual. How do you interpret this element? Are the vertebrae incomplete and the original articular surfaces are missing?

Additional comments

Beside these questions I would suggest that the authors add a brief description of the studied material in the Material & Methods section.
L. 310. Please eliminate the repetition “ The robust centra are robust…”
L. 492. What are the rib fragments that you mention? IWCMS 2018.30.7 and 8 (Rib fragments). This specimens are both fragments of an ilium, is that correct?

---

## Round 0.2 · accepted · Accept

Thank you for your close attention to the comments from the reviewers.